# Mechanically-primed voltage-gated proton channels from angiosperm plants

Chang Zhao [1], Parker D. Webster[1], Alexis De Angeli [2] ✉ & Francesco Tombola [1] ✉

Voltage-gated and mechanically-gated ion channels are distinct classes of membrane proteins that conduct ions across gated pores and are turned on by electrical or mechanical stimuli, respectively. Here, we describe an Hv channel (a.k.a voltage-dependent H[+] channel) from the angiosperm plant *A. thaliana* that gates with a unique modality as it is turned on by an electrical stimulus only after exposure to a mechanical stimulus, a process that we call priming. The channel localizes in the vascular tissue and has homologs in vascular plants. We find that mechanical priming is not required for activation of non-angiosperm Hvs. Guided by AI-generated structural models of plant Hv homologs, we identify a set of residues playing a crucial role in mechanical priming. We propose that Hvs from angiosperm plants require priming because of a network of hydrophilic/charged residues that locks the channels in a silent resting conformation. Mechanical stimuli destabilize the network allowing the conduction pathway to turn on. In contrast to many other channels and receptors, Hv proteins are not thought to possess mechanisms such as inactivation or desensitization. Our findings demonstrate that angiosperm Hv channels are electrically silent until a mechanical stimulation turns on their voltage-dependent activity.

Hv channels are ancient membrane proteins that originated more than 1.5 billion years ago, before plants separated from animals and fungi[1]. They have important roles in the regulation of a variety of pH-dependent cellular processes such as bioluminescence in dinoflagellates[2], biogenic calcification in coccolithophores and corals[3,4], and the production of reactive oxygen species (ROS) by the NADPH oxidase in human and other animals[5–7].

pH regulation is essential for plant growth and development, as well as for the response to abiotic stress and immunity[8–11]. Plants also express multiple respiratory burst oxidase homolog (RBOH) proteins, which produce ROS under different conditions, including bacterial and fungal infections[12–14], and the root response to mechanical stimuli[15]. Some of the constituents of the pH regulation network and the root mechanotransduction pathway have been identified[16–18], but little is known about the contribution of proton channels to these processes. The existence of plant Hv proteins was first proposed more than a

decade ago based on sequence homology with proton channels identified in algae[19], but whether these proteins can function as proton channels has not been demonstrated.

Hv channels are members of a large family of proteins containing voltage-sensing domains (VSDs), which also includes voltage-gated sodium, potassium, and calcium channels[20–23], as well as voltage-sensitive phosphatases[24]. VSDs contain four transmembrane segments, S1 through S4, and their main function is to detect electrical stimuli in the form of changes in membrane potential. In most voltage-gated ion channels the VSD is connected to a distinct pore domain that conducts ions when turned on by the VSD. In Hv channels however, the VSD acts both as a sensor of the electrical stimulus and as a gated conduction pathway for protons[25,26]. Furthermore, the VSD of Hv channels is connected at the C-terminus to a cytoplasmic coiled-coil domain (CCD) that mediates dimerization[27–29]. The S4 segment of the VSD contains periodically

[1]Department of Physiology and Biophysics, University of California, Irvine, CA 92697, USA. [2]IPSiM, University of Montpellier, CNRS, INRAE, Institut Agro, Montpellier, France. ✉e-mail: alexis.deangeli@cnrs.fr; ftombola@uci.edu

arranged positively charged residues (R or K), responsible for sensing the electric field inside the membrane[30,31].

In most voltage-gated channels, the electrical stimulus sensed by the S4 segment is necessary and sufficient to turn on ion conduction. Membrane stretch is known to modulate voltage-gated channels−including Hv channels−in different ways[32–35], but the mechanical stimulus is not required for activation. On the other hand, mechanically-gated channels, such as Piezo proteins and members of the $K_{2P}$, TRP, and Degenerin channel families, require a mechanical stimulus to open[36–38]. Other channels that have been shown to open in response to membrane stretch include cMscS-like (MSL), OSCA, and MCA proteins from plants[39–41]. Here, we find that members of the Hv channel family from the non-flowering plants *Picea sitchensis* and *Selaginella moellendorffii* produce voltage-dependent outward-rectifying proton currents with characteristics similar to those of Hv channels from animals and *Basidiomycota* fungi. In contrast, homologous Hv channels from the angiosperm plants *Arabidopsis thaliana* and *Theobroma cacao* fail to produce measurable currents upon membrane depolarization, but can be rescued from their electrically silent state by membrane stretch in a process we call mechanical priming. Using sequence homology analysis, mutagenesis, and electrophysiological measurements, combined with AI-based structural modeling, we identify a number of residues located at the inner and outer ends of the S4 segment that play an important role in the new gating modality, which requires both mechanical and electrical stimulation. We propose that interactions mediated by these residues lock the channels in a silent resting state, and that the role of the mechanical stimulus is to destabilize these interactions, thus allowing channel opening by the electrical stimulus. In Hv proteins that do not require mechanical priming, the equivalent interactions appear to be less extensive or weaker, resulting in a constitutively destabilized resting state.

## Results

### Differences in channel activity between Hv proteins from flowering and non-flowering plants

Given the importance of pH regulation in plants[8,42], we asked whether Hv proteins that function as proton channels could be found in the model plant *Arabidopsis thaliana*. We identified a homolog of human Hv1 through BLAST search of the *A. thaliana* genome (see "Methods" for sequence ID). The homolog (AtHv1) shares 18% sequence identity with the human channel and has a predicted topological organization typical of other voltage-gated proton channels, with four transmembrane segments forming the VSD and a CCD at the C-terminus (Fig. 1a, b). To confirm the expression and localization pattern of the protein in *Arabidopsis*, we cloned the genomic fragment of AtHv1 (gHv1) including the coding sequence and the promoter region upstream the start codon in chromosome 1. We fused gHv1 with a C-terminal GFP and we used this construct to generate transgenic plants expressing AtHv1-GFP under the control of the native promoter and observed protein expression pattern using confocal fluorescence microscopy (Fig. 1c, e). We found significant expression of AtHv1 in the root vascular tissue in the elongation and differentiation zones (Fig. 1c). The fluorescent signal is likely deriving from protoxylem and the metaxylem cells of the primary xylem (Fig. 1e) and displays properties distinct form the typical auto fluorescence of *Arabidopsis* vasculature (Fig. S1). We then expressed AtHv1 in *Xenopus* oocytes and attempted to record currents from inside-out patches under conditions previously utilized to characterize animal and fungal Hv channels[43,44]. However, we could not detect any obvious

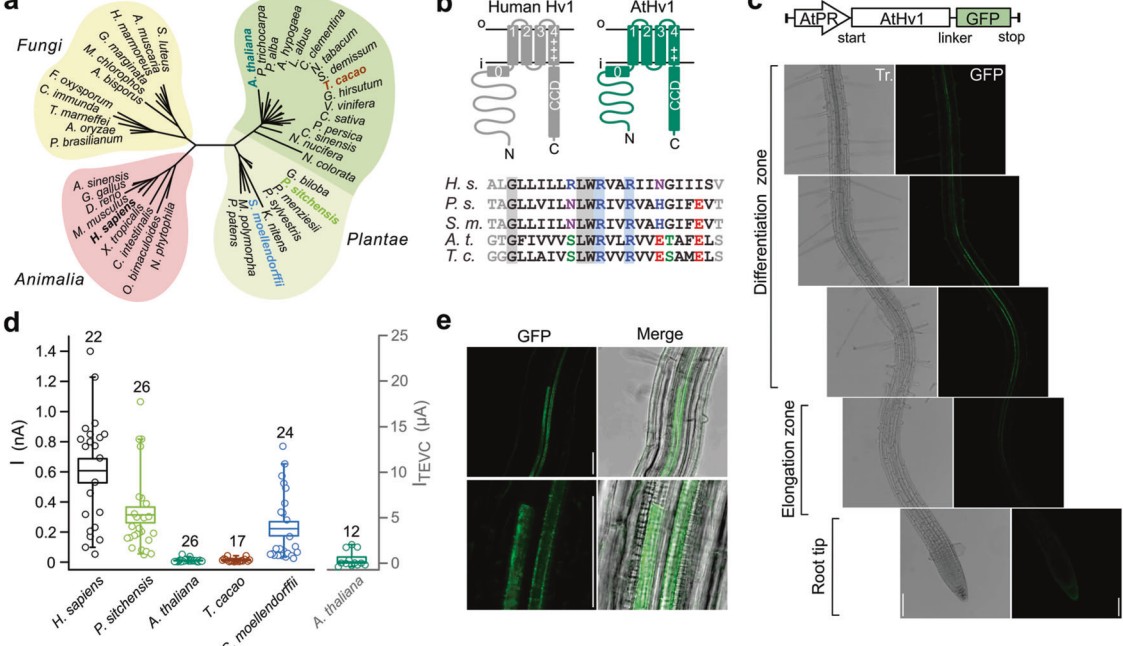

**Fig. 1 | Plant Hv channels: AtHv1 expression in *Arabidopsis* and *Xenopus* oocytes. a** Cladogram showing the relationship between plant Hvs and homologs from animals and fungi. Angiosperm channels, including AtHv1, form a cluster (darker shade) distinct from other plant Hvs. **b** Predicted membrane topology of AtHv1 compared to human Hv1 (H.s.). Alignment of S4 segments of plant Hvs highlights conserved positively charged residues. **c** AtHv1 expression in *Arabidopsis* assessed by confocal fluorescence microscopy. In the imaged transgenic plants, the translation of GFP-tagged protein is controlled by the genomic promoter region AtPR. Strongest signal was detected from the stele (where vascular tissue is located) at the transition between differentiation and elongation zones. Gray-scale images: bright field light. Scale bars: 100 μm. **d** Magnitude of proton currents (*I*) measured

in inside-out patches from *Xenopus* oocytes expressing Hv channels from the indicated species. Test potential: 80 mV, holding potential: −60 mV, $pH_i = pH_o = 6.0$. Current measured in TEVC ($I_{TEVC}$) from cells expressing AtHv1 is shown on the right. Test potential: 80 mV, holding potential: −50 mV. Numbers of biologically independent measurements (*n*) for the individual channels are shown atop box plot. Boxes indicate means ± SE. Whiskers show 5th and 95th percentiles. Source data are provided as a Source Data file. **e** Magnification of the root vasculature in the differentiation zone expressing GFP-tagged AtHv1. Merge panels combine light transmission and confocal images. Scale bars: 50 μm (top and bottom). Confocal images shown in (**c**) and (**e**) are representatives of three distinct transgenic lines (see Fig. S1).

current (Fig. 1d). Suspecting insufficient expression levels, we recorded from whole oocytes using Two-Electrode Voltage Clamp (TEVC), but still the measured current was indistinguishable from background leak (Fig. 1d, see "Methods").

Since the first report of Hv sequences in plants, based on sequence homology with proton channels identified in algae[3], no study has reported channel activity with these proteins. So, we asked whether the lack of AtHv1-mediated currents was an isolated case or a general feature of plant Hvs. A phylogenetic analysis of AtHv1 homologs revealed a significant sequence divergence between Hvs from angiosperms, gymnosperms, and other vascular plants (Figs. S2 and S3), suggesting functional diversity as previously found with fungal Hvs[44]. To test whether representative channels from different plants behaved like AtHv1, we expressed Hv homologs from *Theobroma cacao* (TcHv1), *Picea sitchensis* (PsHv1) and *Salaginella moellendorffii* (SmHv1) in *Xenopus* oocytes and measured channel activity in inside-out patches. TcHv1, an angiosperm channel like AtHv1, and 46% identical to AtHv1, failed to produce detectable currents. However, PsHv1, a gymnosperm channel 33% identical to AtHv1 produced large voltage-dependent currents (Fig. 1d) that resembled those of other known Hv channels (Fig. 2). SmHv1, a representative from ancient vascular plants and 27% identical to AtHv1, produced similar currents compared to PsHv1 (Fig. 1d).

## Plasma membrane localization of AtHv1

We then asked whether the lack of channel activity observed with AtHv1 was the result of lack of trafficking to the plasma membrane. We expressed the hemagglutinin antigen (HA)-tagged channel in *Xenopus* oocytes, treated the cells with a membrane-impermeable biotinylating agent, followed by cell lysis, pulldown of biotinylated proteins, and Western blotting. The procedure was run in parallel with cells expressing HA-tagged PsHv1 for comparison. Because of its larger extracellular region, PsHv1 offers a bigger biotinylation target than AtHv1. Nevertheless, the amounts of pulled-down proteins in the biotinylation assay were similar for AtHv1 and PsHv1 (Fig. S4a, b).

We also wondered whether AtHv1 membrane trafficking could be assessed in *Arabidopsis thaliana* cells using the membrane staining dye FM 4-64. The fluorescence signal of the dye in the plant xylem was very weak (possibly due to low permeability of FM 4-64 through the Casparian strip), precluding the use of GFP-tagged AtHv1 expressed under the native promoter (i.e., gHv1-GFP). So, we cloned the cDNA of AtHv1 from an *Arabidopsis* root cDNA library and we generated transgenic plants ubiquitously expressing the GFP-tagged channel under the Cauliflower Mosaic Virus p35S promoter (Fig. S4a) and observed protein localization in cortical root tip cells with intense FM 4-64 signal. While a small fraction of AtHv1 appeared to be in intracellular compartments, the fluorescence profile of the plasma membrane dye overlapped with the AtHv1 signal (Fig. S4c–e). Taken together, these

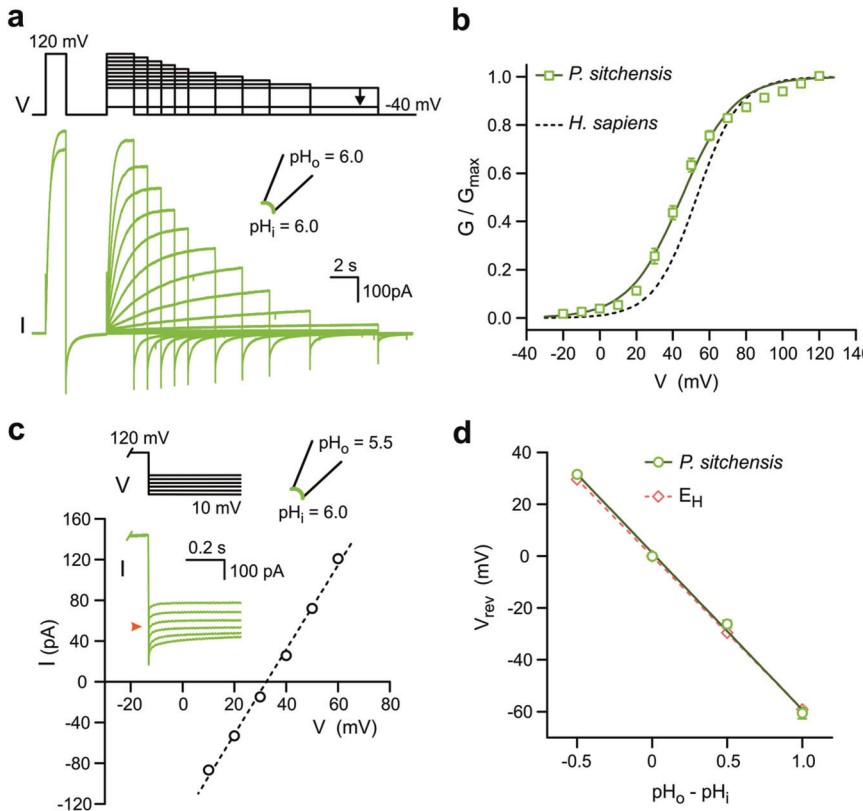

**Fig. 2 | Voltage dependence and proton selectivity of PsHv1. a** Representative current traces measured from membrane patch containing PsHv1. Voltage clamp protocol is shown above the traces. The pre-pulse at 120 mV was used for correction of the current rundown. Only the initial and final currents at the pre-pulse are shown for clarity. **b** $G-V$ relationship of PsHv1 calculated from current traces like those shown in (**a**). Individual points are means of $n = 5$ biologically independent measurements ± SEM. Curve in solid line is derived from Boltzmann fit of the data (see "Methods"). $V_{1/2} = 44.8 \pm 1.6$ mV, slope $= 14.3 \pm 0.3$ mV. $G-V$ curve for hHv1 is shown as reference in dashed line, from ref. 25. **c** Example of measurement of reversal potentials ($V_{rev}$) for PsHv1-mediated current in the presence of a transmembrane pH gradient (ΔpH) of −0.5. Current was measured at the indicated voltages after a depolarization step to 120 mV. Orange arrowhead in inset indicates 0 pA. **d** $V_{rev}$ as a function of ΔpH. Each point is the mean of $n$ biologically independent measurements ± SEM ($n = 4$, 3, 4, 3 for ΔpH = − 0.5, 0, 0.5, 1, respectively). Error bars are not shown where smaller than symbols. Slope of linear fit is $−60 \pm 2$ mV/pH unit. The Nernst potential for protons ($E_H$) is displayed as dashed line (slope: −59 mV/pH unit). ΔpH values of −0.5, 0, 0.5, and 1, correspond to the ($pH_i|pH_o$) pairs (6.5|6.0), (6.0|6.0), (6.0|6.5), and (5.5|6.5), respectively. Source data are provided as a Source Data file.

findings suggest that AtHv1 is trafficked to the plasma membrane but does not respond to membrane depolarization like PsHv1 or SmHv1.

## Comparison between PsHv1 and non-plant Hv channels

We assessed the voltage range of activation of PsHv1 from tail currents measured in inside-out patches as previously done with other Hv channels ($pH_i = pH_o = 6.0$, Fig. 2a). The $V_{1/2}$ of the conductance-vs-voltage relationship ($G−V$) was $45 \pm 2$ mV (Fig. 2b), not far from the $V_{1/2}$ measured with human Hv1, $53 \pm 3$ mV[25], and the fungal channel SlHv1, $46.5 \pm 2.3$ mV[44], under the same conditions. Proton selectivity was determined measuring current reversal potential ($V_{rev}$) under different proton gradients ($\Delta pH$ defined as $pH_o − pH_i$, Fig. 2c). $V_{rev}$ tightly followed the value predicted by the Nernst relationship for protons ($E_H$, Fig. 2d), as observed with all Hv1 homologs studied so far[1].

A key characteristic of most Hv channels is that their voltage range of activation shifts with $\Delta pH$ in a way that ensures channel opening when the $H^+$ electrochemical gradient favors outward proton currents[45]. We asked whether this was the case also for PsHv1, first measuring the current vs. voltage relationship ($I−V$) under symmetrical pH conditions ($\Delta pH = 0$) and then repeating the measurement with a $\Delta pH$ of one unit in either direction (Fig. 3a, b). The $\Delta pH$ was changed by exposing inside-out patches to different bath solutions (change in $pH_i$). To quantify the shifts in voltage range of activation, we measured $V$-threshold ($V_T$), defined as the V-intercept of the linear fit of the steepest segment of the $I−V$ curve for each $\Delta pH$ (Fig. 3a, b). The shifts were $−92$ mV for $\Delta\Delta pH = 1$ and $+96$ mV for $\Delta\Delta pH = −1$ (Fig. 3c). Only outward currents were observed under the tested conditions.

The $V_T$ for PsHv1-mediated currents significantly changed also as a function of $pH_i$ in the absence of a pH gradient ($pH_i = pH_o$) (Fig. 3d). To

estimate how much $V_T$ depends on $\Delta pH$, without the contribution of the dependence on $pH_i$, we can subtract the $V_T$ shift of ~35 mV per $pH_i$ unit (Fig. 3d) from the shifts reported in Fig. 3c, which leads to an average ~60 mV shift per $\Delta pH$ unit.

An alternative method to determine the extent to which the transmembrane pH gradient affects the voltage range of activation of Hv channels is to measure the $G − V$ curves under different $\Delta pH$ conditions and then calculate the shift in $V_{1/2}$ per unit of $\Delta pH$. When we used this approach (Fig. S5a, b) we calculated shifts that were indistinguishable from those determined from $V_T$ values (Fig. S5d). It should be noted here that $V_T$ does not depend on the expression level of the channel or the absolute proton concentration. In fact, it can be shown (Fig. S5c) that $V_T$ is linked to the parameters of the Boltzmann fit of the normalized $G − V$ (see "Methods", Eqs. (1)−(3)).

One structural feature that sets apart plant Hvs from animal, fungal, and even algal homologs is the lack of a highly conserved motif in the S2 transmembrane helix containing a phenylalanine and a negatively charged residue separated by two, usually hydrophobic, residues (FMME in hHv1, Fig. S3). The phenylalanine in particular is considered part of the charge transfer center that interacts with the S4 arginines in most VSDs[46–48], and has been proposed to form the gate in Hv channels[49,50]. The phenylalanine also plays an important role in the inhibition of Hv channels by arginine mimics such as 2-guanidinobenzimidazole (2GBI) and its 5-chloro derivative ClGBI[51,52]. We measured PsHv1-mediated currents in inside out patches before and after perfusion of each compound in the bath compartment. We found that 2GBI and ClGBI inhibited PsHv1 with much lower potency compared to Hv channels containing the S2 motif, such as hHv1 tested under the same conditions (Fig. 3e, f). These results confirm the

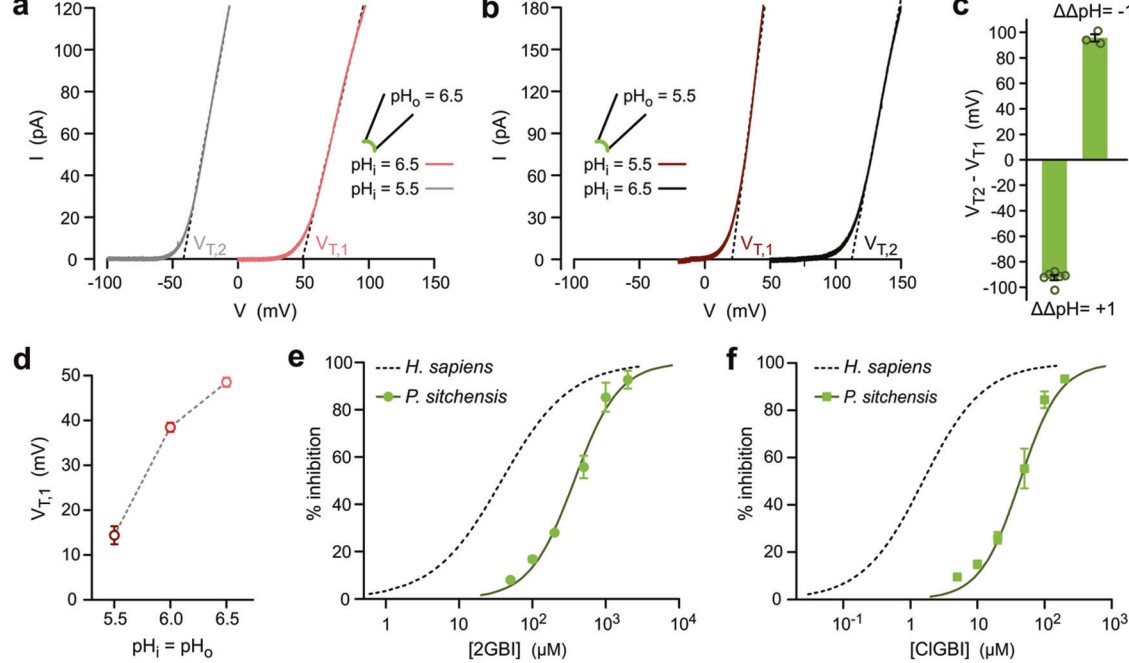

**Fig. 3 | ΔpH dependence of PsHv1 gating and sensitivity to arginine mimics.** Representative $I−V$ relationships for PsHv1 showing shifts in $V_T$ in response to changes in $\Delta pH$ from 0 to 1 (**a**), and from 0 to −1 (**b**). Voltage was changed using ramp protocols described in "Methods". **c** Average shifts in $V_T$ as a function of change in $\Delta pH$ ($\Delta\Delta pH$) measured from $I−V$ s like those shown in (**a, b**). Each bar represents the mean of $n$ biologically independent measurements $\pm$ SEM ($n = 6$ and 3 for $\Delta\Delta pH = 1$ and −1 respectively). **d** $V_{T,1}$ of $I−V$ s from PsHv1 as a function of pH in the absence of gradient ($\Delta pH = 0$). Each point is the mean of $n$ biologically independent measurements ($n = 5, 5,$ and 6 for pH = 5.5, 6.0 and 6.5, respectively). Error bars are SEM. Concentration dependence of inhibition of PsHv1-mediated currents

by arginine-mimic compounds 2GBI (**e**) and ClGBI (**f**). Currents were activated by depolarization to 120 mV in inside out patches ($pH_i = pH_o = 6.0$). The compounds were perfused in the bath solution at the indicated concentrations. Each point is the mean of n biologically independent measurements $\pm$ SD ($n = 5, 4, 4, 4, 4,$ and 3 for 50, 100, 200, 500, 1000, 2000 µM 2GBI, respectively, and $n = 3, 5, 4, 5, 5,$ and 3 for 5, 10, 20, 50, 100, and 200 µM ClGBI, respectively). Hill fits of data points are shown as curved solid lines. Fit parameters are: $IC_{50} = 372 \pm 28$ µM and $h = 1.42 \pm 0.13$ for 2GBI, and $IC_{50} = 39 \pm 8$ µM and $h = 1.46 \pm 0.27$ for ClGBI. Inhibition curves for hHv1 are shown as dashed lines for reference, from ref. 51. Source data are provided as a Source Data file.

importance of the S2 motif for the binding of these arginine mimics within the intracellular vestibule of the VSD.

## Mechanical stimulus rescues activity in Hv channels from angiosperm plants

Because we found AtHv1 natively expressed in vascular cells, we considered the possibility that it might be exposed to mechanical forces associated with the hydraulic pressure maintained by the vascular tissue. Membrane stretch is known to facilitate human Hv1 activation by accelerating channel opening. The channel remains in a facilitated state after the mechanical stimulus is removed and only slowly reverts to the original state[35]. We wondered whether plant Hv proteins could be also mechanosensitive, and whether membrane stretch could rescue the response of AtHv1 and TcHv1 to membrane depolarization. A simple way to assess mechanosensitivity of Hv channels consists in exposing membrane patches to two depolarization pulses, one preceding (A) and one following (B) a transient increase in membrane tension produced by negative pressure on the back of the pipette (Fig. 4a, b). Proton currents are measured at the end of each depolarization pulse ($I_A$ and $I_B$) and the ratio $I_B/I_A$ is calculated (Fig. 4c). If the mechanical stimulus induces an increase in current, then $I_B/I_A > 1$. The $I_B/I_A$ ratio for PsHv1

was ~1.5, similar to the ratio for hHv1, ~1.35[35,44]. SmHv1 was a bit more mechanosensitive with an $I_B/I_A$ of ~2.2, but the largest increases in proton current were observed with AtHv1 and TcHv1, with $I_B/I_A$ ratios of 17 and 13, respectively (Fig. 4c). The appearance of voltage-dependent currents after mechanical stimulation in patches containing Hvs from angiosperms confirms that these proteins reach the plasma membrane, but they are inactive/silent prior to application of membrane stretch.

With AtHv1 and TcHv1, we noticed the appearance of small and slowly-raising currents during pulse A and asked whether lengthening the initial depolarization pulse would affect $I_A$ and consequently the estimate of $I_B/I_A$. We remeasured the response of AtHv1 to mechanical stimulation using a protocol with an initial depolarization (pulse A) three times longer than the one used in Fig. 4b. Despite a small effect on the size of $I_A$, the new ratio $I_B/I_A$ was very similar to the one measured with a 3s-long pulse A (Fig. S6). The appearance of the small currents during pulse A could be due to unintended exposure to membrane stretch before the start of the measurements (e.g., during patch formation).

After mechanical stimulation, AtHv1 remained active for the duration of the measurements, which were usually completed within

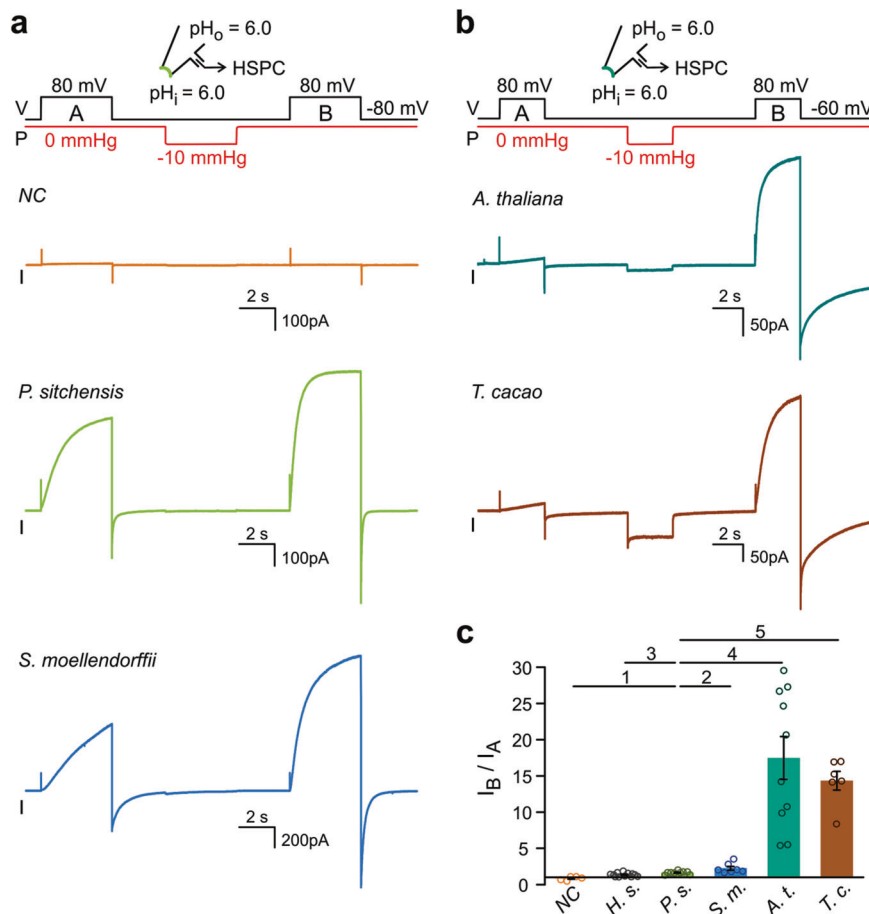

**Fig. 4 | Response of plant Hv channels to mechanical priming.**
**a**, **b** Representative proton currents elicited by membrane depolarization for the indicated plant Hvs, before (step A) and after mechanical stimulus (step B). Membrane stretch was induced via negative pressure ($\Delta P = -10$ mmHg) applied to the patch pipette at resting membrane potential. The duration of the mechanical stimulus was for 4 s in (**a**) and 3 s in (**b**). A current trace from an uninjected oocyte is shown as representative negative control (*NC*). **c** Averaged increases in current caused by the mechanical stimulus. Current values $I_A$ and $I_B$ were measured at the end of depolarization steps A and B, respectively. Bars represent means of $I_B/I_A$ from *n* biologically independent measurements ± SEM, $n = 5$ (*NC*), 12 (*H.s.*), 9 (*P.s.*), 7

(*S.m.*), 10 (*A.t.*), and 6 (*T.c.*). Values for human Hv1 (*H.s.*) are from ref. 44. One-way Welch's ANOVA with Dunnet's T3 test was used for multiple comparison analysis, using PsHv1 as reference. *p* values were: 0.0037 (1), 0.2282 (2), 0.0590 (3), 0.0022 (4), and 0.0008 (5). Source data are provided as a Source Data file. Inward current was observed during the application of the negative pressure pulse with TcHv1, and to a lesser extent with other plant Hvs, but not with animal or fungal Hvs. The origin of the inward current remains to be determined as there was no correlation between its size of and the number of channels present in the patch (evaluated by the size of the current elicited by membrane depolarization after priming).

5–10 min. The return to the silent state after priming could not be followed over time due to superimposing rundown of the current, which was observed in AtHv1 and the other plant proton channels and resembled the equivalent process previously reported in human Hv1[25].

## Characteristics of AtHv1 and SmHv1-mediated currents

Once AtHv1 channels were exposed to membrane stretch, their voltage-dependent current remained elevated long enough to allow basic characterization. Membrane stretch was also used to accelerate the activation of SmHv1. We assessed proton selectivity of AtHv1 and SmHv1 post-mechanical stimulus by measuring current reversal potentials under different proton gradients (Fig. S7). The results were similar to those obtained with unstimulated PsHv1. $V_{rev}$ closely followed the Nernst potential for H[+] even though the measurements were performed in the presence of symmetrical chloride concentrations ($[Cl^-]_i = [Cl^-]_o = 5$ mM), indicating that chloride permeability was negligible. Because proton selectivity was evaluated in the absence of monovalent metal cations, such as K[+] or Na[+], we separately tested whether these ions can permeate to a significant extent. After channel activation in inside-out patch, we measured the current reversal potential in the presence of 1 mM K[+] or Na[+] in both intra- and extracellular solutions ($V_{rev,1}$). We then increased the concentration of the metal ion 10-fold in the intracellular solution and measured the reversal potential again ($V_{rev,2}$), while keeping symmetrical pH conditions ($pH_i = pH_o = 6.0$) (Fig. S8a). The resulting $\Delta V_{rev}$ ($V_{rev,2} - V_{rev,1}$) were compared to the values expected for different relative permeabilities ($P_{K^+}/P_{H^+}$ and $P_{Na^+}/P_{H^+}$) calculated with the Goldman-Hodgkin-Katz voltage equation (Eq. (5)) under the assumption of negligible Cl[-] permeability. The $\Delta V_{rev}$ measured were consistent with $P_{K^+}/P_{H^+}$ and $P_{Na^+}/P_{H^+}$ of $10^{-5}$ or lower (Fig. S8b).

We next investigated the $\Delta pH$ dependence of the voltage range of activation of AtHv1 and SmHv1 by measuring their $I-V$ relationships and $V_T$ values as a function $\Delta pH$ (Fig. S9a–f). $V_T$ shifted of 78–93 mV per unit of $\Delta pH$ for AtHv1 and ~76 mV for SmHv1 (Fig. S9h). As observed with PsHv1, $V_T$ also changed in response to pH in the absence of a pH gradient ($pH_i = pH_o$). The shift in $V_T$ for AtHv1 was ~38 mV per pH unit (Fig. S9c), but was considerably smaller for SmHv1, ~9 mV per pH unit (Fig. S9f). Once corrected for the shifts under symmetrical pH, the $V_T$ shifts per unit of $\Delta pH$ were 40–53 mV and ~67 mV for AtHv1 and SmHv1, respectively. Overall, the functional characteristics of AtHv1 and SmHv1 post-mechanical stimulation resembled those of unstimulated PsHv1, with one exception, the appearance of inward currents (Fig. S9b–d). For example, at $pH_i = pH_o = 5.5$, AtHv1 started opening when the electrochemical gradient favored proton influx (Fig. S9b). This is better appreciated in Fig. S9g, which shows that the conductance of AtHv1 is ~30% of $G_{max}$ at $V_m = E_H = 0$ mV.

Finally, we tested the effects of 2GBI and ClGBI on AtHv1- and SmHv1-mediated currents post-mechanical stimulation (Fig. S9i). Despite some small differences in the extent of inhibition by 2GBI, AtHv1, SmHv1, and PsHv1 shared a low sensitivity to the compounds, which differentiated them from all other Hv channels that possess the charge transfer center phenylalanine in S2.

## Protein regions involved in AtHv1 response to membrane stretch

Because membrane stretch appeared to convert AtHv1 from an inactive/silent state to a state in which it responds to electrical stimulation, we named the process causing the conversion "mechanical priming" and set out to identify its molecular determinants. We designed a number of chimeric channels in which different parts of PsHv1 were swapped with the corresponding regions of AtHv1 and asked whether any of these chimeras required mechanical priming in order to be activated by membrane depolarization (Fig. 5a). Each chimera was examined with the same two-pulse protocol used to quantify the response of wild type AtHv1 and TcHv1 to membrane stretch (Fig. 4b).

$I_B/I_A$ ratios were calculated and compared to the ratios for the parent proteins PsHv1 and AtHv1 (Fig. 5b–d). We started by exchanging non-transmembrane regions with the highest sequence divergence. The swaps included the extracellular regions connecting segment S1 to S2 (chimera ChE1-2), the intracellular region connecting segment S2 to S3 (chimeras ChI2-3a and ChI2-3b) and the cytoplasmic N- and C-terminal domains (chimera ChNC). All the resulting hybrid channels retained a mild mechanosensitivity comparable to PsHv1, with $I_B/I_A$ ratios in the range 1.5–2.6 (Fig. 5b–d), indicating that the swapped regions are unlikely to play an important role in AtHv1 mechanical priming.

We then turned to the transmembrane helices forming the VSD and focused on the S4 segment because of its critical role in voltage-dependent gating. We transferred the S4 helix and part of the extracellular loop connecting it to S3 from AtHv1 to PsHv1 and found that the resulting chimera (ChE3-4.S4, Fig. 5a) did not produce measurable currents with or without mechanical stimulation (Fig. 5c). The $I_B/I_A$ ratio for this channel (~1.2, Fig. 5d) mostly reflected the properties of background leak current (Fig. 4c, NC). We noticed that AtHv1 and other Hvs from flowering plants contained a negatively-charged glutamate close to the inner end of S4 (E173) and concurrently a positively-charged lysine close to the inner end of S2 (K117), while the residues at the corresponding positions in PsHv1 and other Hvs of non-flowering plants were a histidine (H208) and an asparagine (N152), respectively (Fig. S3). The ChE3-4.S4 channel only contained the S4 glutamate from AtHv1, but still contained the neutral asparagine in S2, so we mutated it to lysine (N152K) to replicate the charged pair found in AtHv1. The resulting chimera ChE3-4.S4.K (Fig. 5a) produced measurable currents after mechanical stimulation like AtHv1, with an $I_B/I_A$ ratio of ~26 (Fig. 5c, d). The result indicates that the swapped parts in the chimera play an important role in preventing the channel from activating in the absence of membrane stretch.

## Searching for residues responsible for mechanical priming in structural models of AtHv1 and PsHv1

Structural information on the VSDs of plant Hvs could be obtained by homology modeling based on partial structures of animal channels[26,53,54]. However, this approach is complicated by the large sequence divergence between plant and animal Hvs and between Hvs from flowering and non-flowering plants. Therefore, we used the AI-based prediction algorithm AlphaFold2[55,56] to generate structural models of AtHv1 and PsHv1 (Fig. S10) and searched for differences between the two structures that could point to specific molecular determinants of mechanical priming. The positions of the selectivity filter in S1 and the positively-charged residues in S4 that sense the electrical field were consistent with the structures of animal Hv channels[26,54]. Because AtHv1 and PsHv1 only open in the presence of membrane depolarization, we assumed that the modeled VSDs represented a closed state. The most apparent difference between the two models was in the extracellular portion of the VSDs. In PsHv1, the long S1 helix protruded from the extracellular side of the transmembrane region and was connected to the S2 helix via an extended loop (Fig. S10b). The feature was absent in AtHv1, consistent with its much shorter S1–S2 loop (Fig. S10a). However, based on the lack of strong mechanical priming in the ChE1-2 chimera (Fig. 5c, d), this structural feature was not studied in further detail.

We then turned to the portion of the VSD that was swapped in the ChE3-4.S4K chimera because the resulting channel was strongly mechanically primed. Two major sets of residues in S4 and its surroundings appeared to be different between the two models. One set, located in the intracellular vestibule of the proton conduction pathway below the selectivity filter, comprised a ring-shaped network (RSN) of interacting hydrophilic/charged residues (Fig. 6a) nested within a hydrophobic region (gray bands in helices S1 through S4, Fig. 6a). In AtHv1, the network included K117 in S2, as well as E173 and T174 in S4 (KET residues). In PsHv1, it included N152

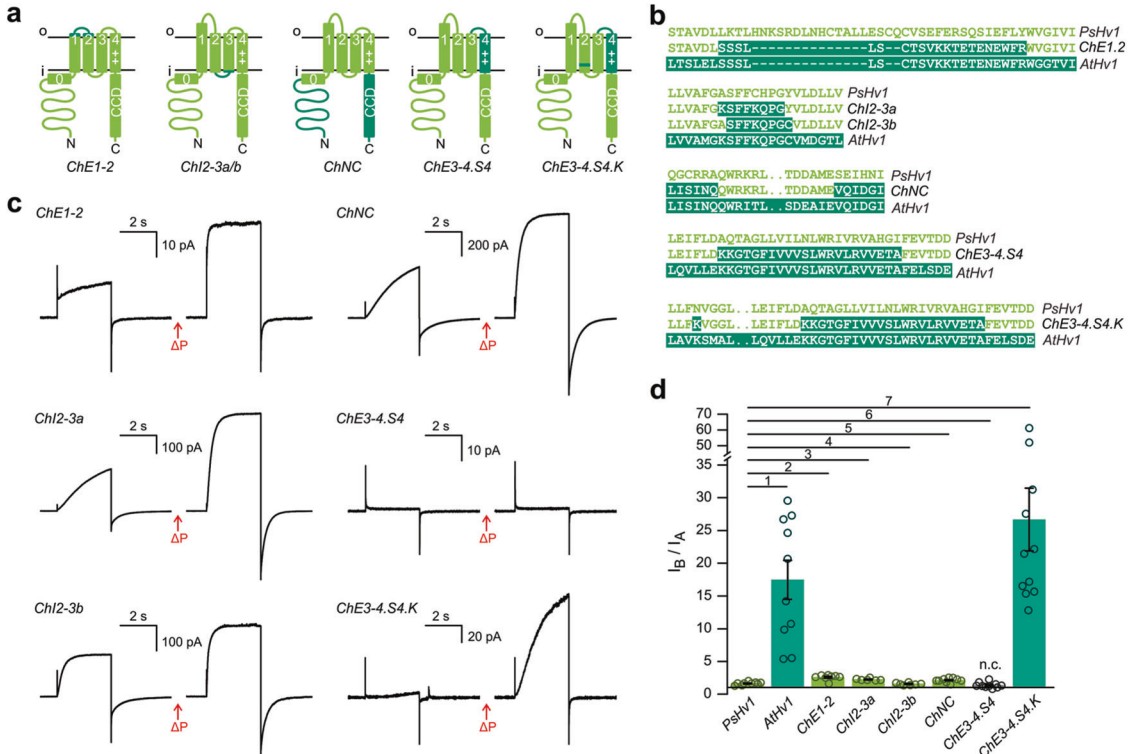

**Fig. 5 | Identification of the protein regions responsible for mechanically primed AtHv1 activation.** Schematics of chimeric proteins with swaps between PsHv1 and AtHv1 (**a**), and sequence details (**b**). **c** Representative proton currents elicited by membrane depolarization for the indicated chimeras and mutants, before and after mechanical stimulus ($\Delta P$). Stimulation protocol as in Fig. 4b. **d** Averaged increases in current caused by the mechanical stimulus. $I_A$ and $I_B$ measured as in Fig. 4c. Bars are means of $I_B/I_A$ from $n$ biologically independent measurements $\pm$ SEM, $n = 9$ (PsHv1), 10 (AtHv1), 8 (ChE1-2), 6 (CHI2-3a), 7 (CHI2-3b), 10 (ChNC), 10 (ChE3-4.S4), and 11 (ChE3-4.S4.K). One-way Welch's ANOVA with Dunnet's T3 test was used for multiple comparison analysis, using PsHv1 as reference. $p$ values were: 0.0030 (1), 0.0008 (2), 0.0003 (3), 0.9363 (4), 0.0093 (5), 0.4273 (6) and 0.0026 (7). Expression of ChE3-4.S4 did not result in measurable currents (n.c.). The $I_B/I_A$ ratio for this chimera reflects the mechanosensitivity of the background leak current. Source data are provided as a Source Data file.

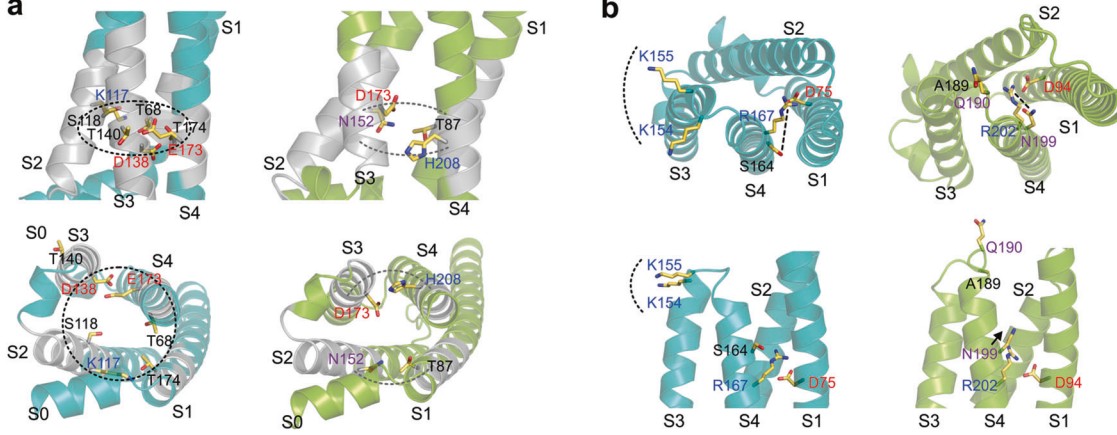

**Fig. 6 | Main differences in AI-derived structural models of AtHv1 and PsHv1 in the regions involved in mechanical priming.** **a** Models of the VSDs of AtHv1 (teal) and PsHv1 (yellow-green): side view (top panel) and view from inner side of membrane (lower panel). Models compare ring-shaped networks (RSN) of hydrophilic residues in the intracellular vestibules. Hydrophobic helical regions surrounding the networks are colored in gray. Dotted curved lines indicate the extensions of the networks. **b** Same models as in (**a**) comparing residues in the extracellular portions of S4 and the S3–S4 loop. View from outer side of membrane (top panel), side view (lower panel). Dotted curved lines mark positively charged residues in AtHv1 facing the membrane outer surface that are missing in PsHv1. Dashed lines represent the distance between the first S4 arginine and the hydrophilic residue at position n-3 (S164 in AtHv1, N199 in PsHv1). The distance is larger in AtHv1 because S164 points away from the center of the VSD. In PsHv1, N199 points toward the center of the VSD (black arrow) where R202 is located.

(S2) and H208 (S4). The AtHv1 RSN appeared to encircle the interior of the VSD, while the PsHv1 network was less extensive and split in two patches, one between S1 and S2 and the other between S3 and S4 (Fig. 6a). The second set of residues located on the outer side of the VSD included K154 and K155 (S3–S4 loop), and S164 (S4) in

AtHv1, which correspond to A189, Q190, and N199 in PsHv1, respectively (Fig. 6b). The residues in the two structural models were not just different in nature but also in orientations. The two lysines in AtHv1 faced the membrane surface, whereas the corresponding residues in PsHv1 faced the extracellular medium. S164 in

AtHv1 pointed away from the first S4 arginine, while N199 in PsHv1 pointed toward it (Fig. 6b).

### Testing molecular determinants of mechanical priming

To determine whether the KET residues in the RSN of AtHv1 are involved in mechanical priming, we generated the PsHv1.KET mutant containing the amino acid substitutions N152K, H208E, and G209T (Fig. 7a, b), and assessed its response to membrane depolarization before and after mechanical stimulation (Fig. 7c). The $I_B/I_A$ ratio of this mutant (6.5 ± 0.6, Fig. 7d) was significantly higher than the corresponding ratio of the parent PsHv1 protein (1.5 ± 0.1), indicating that the KET residues were able to enhance mechanical priming in PsHv1.

We also tested the residues on the outer end of the VSD, generating the ChE3.4 construct in which the part of the AtHv1 S3–S4 loop containing K154 and K155 is transplanted into PsHv1, and the PsHv1.S construct which carries the N199S substitution (Fig. 7a, b). Both channels showed enhanced mechanical priming with an $I_B/I_A$ ratio of 7.2 ± 0.5 for ChE3.4, and 3.3 ± 0.2 for PsHv1.S (Fig. 7c, d). When we combined the two modifications together in the ChE3.4.S construct, we found an even stronger enhancement of mechanical priming with an $I_B/I_A$ ratio of 27 ± 4 (Fig. 7c, d).

From these results, we conclude that the residues forming the RSN and the residues located at the top of the S4 segment (K154, K155, and S164) are major determinants of mechanical priming. The KET mutations increased the response to mechanical stimulation mostly by increasing the steady-state current at the end of the test depolarization (stimulus B) compared to the control depolarization (stimulus A) (Fig. 7c). The ChE3.4.S mutations, on the other hand, made the requirement for mechanical priming more stringent by dramatically slowing down voltage-dependent activation (Fig. 7c). These effects suggest that the KET and ChE3.4.S mutations alter the energy barrier separating the channel open state from the closed state in different ways. Nonetheless, both types of mutations appear to contribute to the lowering of the energy barrier after mechanical stimulation.

## Discussion

Most ion channels are prevented from opening under the wrong conditions or from staying open for too long by multiple mechanisms, including inactivation and desensitization. Hv channels are not known to inactivate or desensitize, as a result, other regulatory mechanisms are needed to guard against overactivation. In animals, excessive Hv channel activity can lead to elevated production of ROS and imbalances in intra- or extracellular pH, with severe pathological consequences, including neuroinflammation[57–59] and cancer[60,61]. In plant cells, the membrane potential is maintained by the activity of plasma membrane proton pumps and requires tight control of the pH gradient. The consequences of Hv channel overactivation in plants remain to be investigated, but the process of mechanical priming described here appears to restrict channel opening by membrane depolarization to conditions in which the membrane has been exposed to a mechanical stimulus.

The S4 helix of animal and fungal Hvs contains the characteristic sequence motif RLWRXX(R/K), where X represents a hydrophobic amino acid. The three highly conserved positively charged residues in the motif play a key role in voltage sensing[30,31]. The corresponding S4 motif of plant Hvs is (S/N)LWRXXR (Fig. 1b). The lack of the first arginine suggested a weaker voltage dependence of activation of plant channels compared to animal and fungal homologs, which is in agreement with the finding that the G–V curves of plant Hvs are less steep (larger slope parameters) than the G–V curves of animal and fungal homologs measured under the same conditions (e.g., G–V slopes for Hvs from *P. sitchensis*, *H. sapiens*, and *A. oryzae* are 14.3 ± 0.3 mV, 11.6 ± 0.6 mV[25], and 7.4 ± 0.9 mV[44], respectively at $pH_i = pH_o = 6.0$). Hv homologs missing the first S4 arginine have been recently identified in *Aplysia* and named Hv2 and Hv3[62]. The voltage dependence of these proteins was also found to be weaker than Hv1 channels.

The voltage range of activation of most animal Hvs shifts ~40 mV per ΔpH unit[45]. For some fungal Hvs, the shifts are larger, 60–85 mV per ΔpH unit[44]. Multiple lines of evidence point to a direct role of the S4 helix in sensing ΔpH via electrostatic interactions[63–65]. The lack of

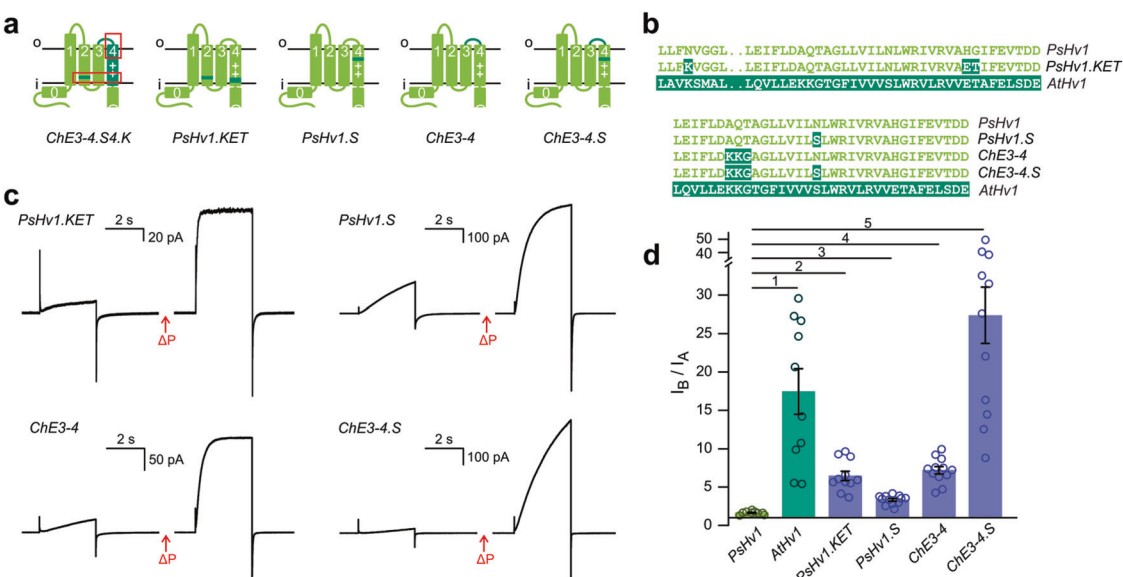

**Fig. 7 | Dissecting the molecular contributors to mechanical priming.** Schematics of PsHv1 alterations at positions suggested by the structural models of AtHv1 and PsHv1 (**a**), and sequence details (**b**). **c** Representative proton currents elicited by membrane depolarization for the indicated chimeras and mutants, before and after mechanical stimulus (ΔP). Stimulation protocol as in Fig. 4b. **d** Averaged increases in current caused by the mechanical stimulus. $I_A$ and $I_B$ measured as in Fig. 4c. Bars are means of $I_B/I_A$ from $n$ biologically independent measurements ± SEM, $n = 9$ (PsHv1), 10 (AtHv1), 11 (PsHv1.KET), 12 (PsHv1.S), 12 (ChE3-4), and 12 (ChE3-4.S). One-way Welch's ANOVA with Dunnet's T3 test was used for multiple comparison analysis, using PsHv1 as reference. $p$ values were: 0.0022 (1), <0.0001 (2–4), and 0.0001 (5). Source data are provided as a Source Data file.

the first charged residue on the S4 motif of plant Hvs raised the possibility that the ΔpH dependence of these channels could be substantially different compared to other Hvs. However, this was not the case, as the voltage range of activation of PsHv1, AtHv1, and SmHv1 shifted 40–67 mV per ΔpH unit (values corrected for pH dependence under symmetrical conditions). These observations suggest the existence of redundancies in the mechanism of ΔpH sensing in Hv proteins.

The difference in mechanosensitivity among plant Hv channels reflects the phylogenetic separation between the proteins from flowering and non-flowering plants (Figs. 4 and S2) and opens the possibility that Hv channels of angiosperms could be regulated in a different way compared to other vascular plants. Sequence homology analysis and structural modeling of PsHv1 and AtHv1 show that the VSDs of the two proteins differ in the E1-2 region (outer end of S1 and the loop connecting it to S2, Figs. 5 and S10). In fungal Hv channels, that same region was found to strongly modulate channel opening and to be a major contributor to the difference in voltage range of activation between Hvs from *Basidiomycota* and *Ascomycota* fungi[44]. However, the E1-2 region does not appear to play an important role in the response of AtHv1 to mechanical stimulation (Fig. 5). Instead, we found that residues located at the inner and outer ends of the S4 segment are major determinants of mechanical priming (Fig. 7).

Upon membrane depolarization, the S4 segment undergoes an outward-movement that drives the channel from a resting/closed conformation (RC) to an activated/open conformation (AO) via at least one activated/closed intermediate (AC)[45,66–68]. Because the KET residues of AtHv1 (K117, E173, T174), when transferred to PsHv1, made the resulting channel more difficult to open in the absence of membrane stretch, we propose that the KET-containing RSN provides extra stabilization to the resting/closed conformation of AtHv1, preventing channel opening (Fig. 8a, b), and that the network weakens as a result of mechanical priming. In other words, the stabilized closed state (RC_S) is converted by the mechanical stimulus into a less stable RC state from which the channel can be opened by membrane depolarization (transitions to AC and AO states, Fig. 8). In PsHv1, the RSN is less extensive

so it could be constitutively weaker, making mechanical priming unnecessary for channel opening (Fig. 8c, d).

The positions of residues K154 and K155 in the AtHv1 model of Fig. 6b suggest that these two positively charged residues could further stabilize the RC_S state via electrostatic interactions with phosphate groups located on the outer surface of the phospholipid membrane. In PsHv1, the corresponding residues are not positively charged and do not point toward the membrane surface (Fig. 6b). As a result, they would not be able to stabilize the RC_S state to a comparable extent.

High-resolution structures of AtHv1 and PsHv1 will be required to confirm the proposed roles for the identified residues in channel activation and the effect of the swap of the S164/N199 residues. Nevertheless, we note here that in the PsHv1 model of Fig. 6b, the first S4 arginine (R202) appears to form a salt bridge with the highly conserved aspartate D94 (part of the selectivity filter[69,70]). In the transition to the activated state, the salt bridge is expected to break, so that S4 can move upward. Residue N199, located one helical turn above R202, seems to interact with the arginine as well (Fig. 6b). The interaction between N199 and R202 could facilitate the transition to the activated state by weakening the R202–D94 salt bridge (Fig. 8b). The corresponding S4 arginine and aspartate in AtHv1 (R167 and D75) are also close enough in space to form a salt bridge (Fig. 6b). In this case however, residue S164 (corresponding to N199), points away from R167 and may not be able to facilitate the transition to the activated state, explaining why transferring S164 to PsHv1 makes the resulting channel harder to open (Fig. 7, PsHv1.S, ChE3-4.S).

As for how membrane stretch can lead to the destabilization of the resting closed state (RC_S → RC transition), we envision two main possibilities: (1) a direct effect caused by mechanical force exerted on the protein through the membrane, or (2) an indirect effect caused by a change in the lipid microenvironment triggered by membrane stretch (Fig. S11a). The lipid-mediated mechanism has been previously proposed for other mechanosensitive channels[71,72]. Furthermore, once the lipid composition around the channels is altered, it may take a relatively long time to revert to its initial state, providing a simple

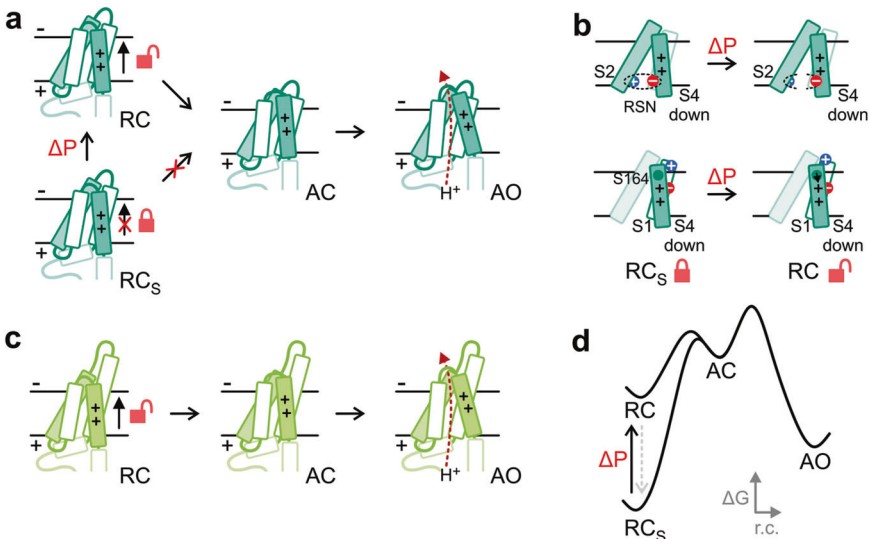

**Fig. 8 | Proposed mechanism of mechanically-primed voltage-dependent gating. a** Activation diagram for AtHv1 shows that the channel is locked in a stabilized resting-closed conformation (RC_S silent state) which prevents channel opening. Mechanical priming destabilizes the resting-closed conformation (transition to RC) allowing the VSD to reach the activated open conformation. **b** Schematics of changes proposed for the RC_S → RC transition, which include the destabilization of the RSN in the inner vestibule (upper panel) and a disengagement of K154 from the membrane surface (lower panel). In addition, S164 could move closer to the first S4

arginine (black arrow) interfering with its electrostatic interaction with D75. The result of these changes is expected to make it easier for S4 to reach the activated conformation upon membrane depolarization. **c** Activation diagram for PsHv1 showing that the channel is in a state similar to the mechanically-primed resting state of AtHv1 (RC) even in the absence of a mechanical stimulus. **d** Reaction coordinate diagram for the process of channel activation showing the proposed change in energy profile induced by mechanical priming. Intermediate states preceding the activated state have been omitted for simplicity.

explanation for why AtHv1 remains primed after the mechanical stimulus is gone. In plants too, the lipid composition of the plasma membrane can be modified by different stimuli[73–75].

Membrane stretch is known to facilitate the activation of the human Hv1 channel[35]. This process was proposed to contribute to brain damage after ischemic stroke by potentiating proton currents in microglial cells in response to cell swelling, leading to excessive ROS production[35,76]. The mechanism underlying the facilitation of human Hv1 is unknown, but it is believed to involve a transition from a state with normal activity to a state with enhanced activity induced by the mechanical stimulus. The present work suggests a general mechanism of mechanosensitivity that agrees with this idea. Plants cells experience mechanical stresses induced by the environment and during the development of tissues and organs[77–79]. The cell-wall and the plasma membrane are the two major actors for the perception and the responses to mechanical stimulations[80]. Mechanosensitive channels are present in the plasma membrane of plant cells[81] and the cell wall composition is modified by mechanical stimulation[82]. In the root vasculature, where AtHv1 is expressed, the composition of the cell wall is modified during the elongation and secondary cell wall deposition (SCW) phases, resulting in changes of extensibility and stiffness[83]. Recent results show that anisotropic stress distribution in the root vasculature defines the patterning of vascular cells controlling their development[84,85]. It is known that during elongation and SCW there is a production of ROS, and AtHv1 could participate to regulate such oxidative bursts[86], similarly to its mammalian homolog in phagocytic cells[1]. In this context, the mechanical forces experienced by the plasma membrane in root vascular cells could activate or deactivate AtHv1. Because in plant cells the membrane potential is largely dependent on the pH gradient[87,88], the activation of AtHv1 could cause changes in membrane potential and consequently act on intracellular signaling. Of particular interest is the signaling involved in auxin-dependent cell growth given its known dependence on H⁺ transport across the plasma membrane[89].

VSD configurations similar to the silent state observed in AtHv1 might exist in other Hv proteins. These states may be sufficiently stable to inhibit channel activation by membrane depolarization but not stable enough to prevent opening altogether. The mechanical stimulus could then destabilize these states, as it is proposed for AtHv1, resulting in a facilitated opening. Thus, what distinguish Hv channels that require mechanical priming from Hv proteins that are potentiated by membrane stretch could be the free energy difference separating the silent state from the primed closed state (Fig. S11b). Future studies will determine whether the resting states of other Hv channels are stabilized by networks of interacting residues similar to those found in plant Hvs. The activation of human Hv1 has been shown to be affected by membrane-bound glycosaminoglycans[90] and cholesterol[91]. The possibility that the effects of these membrane components are related to the process of mechanical priming described here warrants further investigation.

## Methods
### Analysis of protein sequences
Cladogram and phylogenetic tree were constructed with iTOL 5.6.2[92]. Multiple sequence alignment and phylogenetic analysis were performed using Clustal Omega (EMBL-EBI)[93]. Primary sequences of AtHv1, PsHv1, SmHv1, and TcHv1 were further analyzed with MPEx[94] and Coils–ExPASy[95].

**Plant Hv sequences.** *Arabidopsis thaliana* (NP_001321473.1, 236 a.a.); *Arachis hypogaea* (QHO09623.1, 234 a.a.); *Camellia sinensis* (XP_028062609.1, 229 a.a.); *Cannabis sativa* (XP_030507068.1, 278 a.a.); *Citrus clementina* (XP_006436991.1, 250 a.a.); *Gossypium hirsutum* (XP_016748187.2, 248 a.a.); *Klebsormidium nitens* (GAQ80331.1, 289 a.a.); *Lupinus albus* (KAE9604961.1, 239 a.a.); *Marchantia polymorpha*

(PTQ41489.1, 261 a.a.); *Nelumbo nucifera* (XP_010244071.1, 240 a.a.); *Nicotiana tabacum* (XP_016501726.1, 237 a.a.); *Nymphaea colorata* (XP_031481790.1, 236 a.a.); *Nymphaea thermarum* (KAF3773684.1, 237 a.a.); *Physcomitrella patens* (XP_024364004.1, 285 a.a.); *Picea sitchensis* (ABR16431.1, 260 a.a.); *Populus trichocarpa* (XP_006375506.1, 253 a.a.); *Prosopis alba* (XP_028759754.1, 246 a.a.); *Prunus persica* (XP_007225522.1, 265 a.a.); *Selaginella moellendorffii* (EFJ10096.1, 242 a.a.); *Solanum demissum* (ANJ02807.1, 232 a.a.); *Spatholobus suberectus* (TKY64948.1, 223 a.a.); *Theobroma cacao* (XP_017974731.1, 234 a.a.); *Vitis vinifera* (XP_002265639.1, 239 a.a.);

Gymnosperm Hvs from Gymno PLAZA 1.0: *Ginkgo Biloba* (GBI00022260, 261 a.a.); *Picea abies* (PAB00039225, 232 a.a.); *Picea glauca* (PGL00010571, 260 a.a.); *Pinus pinaster* (PPI00014087, 263 a.a.); *Pinus sylvestris* (PSY00020417, 259 a.a.); *Pseudotsuga menziesii* (PME00027381, 261 a.a.).

**Fungal Hv sequences.** *Agaricus bisporus* (XP_007326257.1, 183 a.a.); *Amanita muscaria* (KIL69657.1, 218 a.a.); *Aspergillus flavus* (XP_002381556.1, 211 a.a.); *Aspergillus oryzae* (XP_001825565.1, 211 a.a.); *Cladophialophora immunda* (XP_016251813.1, 259 a.a.); *Fusarium oxysporum* (XP_031056756.1, 230 a.a.); *Galerina marginata* (KDR81513.1, 217 a.a.); *Hypsizygus marmoreus* (RDB21275.1, 215 a.a.); *Mycena chlorophos* (GAT47218.1, 202 a.a.); *Penicillium brasilianum* (CEJ60805.1, 205 a.a.); *Piriformospora indica* (CCA68166.1, 210 a.a.); *Psilocybe cyanescens* (PPQ83343.1, 214 a.a.); *Rhodotorula toruloides* (EGU12623.1, 262 a.a.); *Scleroderma citrinum* (KIM55885.1, 225 a.a.); *Sclerotinia sclerotiorum* (XP_001595616.1, 226 a.a.); *Suillus luteus* (KIK49332.1, 223 a.a.); *Talaromyces marneffei* (EEA28233.1, 309 a.a.).

**Unranked (Protozoa).** *Heterostelium album* PN500 (XP_020427815.1, 280 a.a.).

**Animal Hv sequences.** *Alligator sinensis* (XP_006015244.1, 239 a.a.); *Ciona intestinalis* (NP_001071937.1, 342 a.a.); *Danio rerio* (NP_001002346.1, 235 a.a.); *Gallus gallus* (NP_001025834.1, 235 a.a.); *Homo sapiens* (NP_001035196.1, 273 a.a.); *Mus musculus* (NP_001035954.1, 269 a.a.); *Nicoletia phytophile* (AMK01488.1, 239 a.a.); *Octopus bimaculoides* (XP_014789275.1, 348 a.a.); *Xenopus tropicalis* (NP_001011262.1, 230 a.a.).

### Cloning of the genomic fragment of AtHv1 from *Arabidopsis thaliana*
The genomic fragment of AtHv1 was cloned from *Arabidopsis* on the corresponding gene (At1g10800.2) located on chromosome 1 with a 1408 bp-long promoter region (AtPR). The total length of the AtHv1 genomic fragment was 2119 bp (Ch1: 3598158..3560277). The following primers were used:

5′-<u>GGGGACAAGTTTGTACAAAAAAGCAGGCTTC</u> CAAAGAACCCC AGTCCCA-3′ (forward)

5′-<u>GGGGACCACTTTGTACAAGAAAGCTGGGTG</u> TGGTTTCACAAA TGGAATAT (reverse)

The underlined sequences correspond to the recombination sites attB1 and attB2. The PCR fragment was cloned into pDONR201 by BP reaction. After confirmation by sequencing the coding sequences were cloned into the destination vector pMOP.

### *Arabidopsis* expressing GFP-tagged AtHv1
**Plant material and growth conditions.** For all experiments involving plants, Arabidopsis thaliana (ecotype Columbia) was used. For microscopy observations seeds were kept in the dark at 4 °C for 2 days for stratification and then grown for 5–7 days at 21 °C under 16 h light/8 dark vertically on solid ½ Murashige-Skoog medium (MS).

To generate the transgenic plants overexpressing AtHV1-GFP the coding sequence of AtHv1 without the stop codon was amplified by PCR using the following combination of primers:

5'-<u>GGGGACAAGTTTGTACAAAAAAGCAGGCTTC</u>atgaacatcatcaa-caccgg-3' (forward)

5'-<u>GGGGACCACTTTGTACAAGAAAGCTGGGTGCTA</u>TGGTTTCA-CAAATGGAA-3' (reverse)

The underlined sequences correspond to the recombination sites attB1 and attB2. AtHv1 PCR fragment was cloned into pDONR201 by BP reaction. After confirmation by sequencing the coding sequences were cloned into the destination vector pMDC34 to obtain AtHV1-GFP. The constructs were introduced into Agrobacterium strain GV3101. Transgenic Arabidopsis (Columbia ecotype) were transformed by floral-dip method.

**Imaging and localization.** Confocal pictures were acquired using a Leica SP8 confocal microscope. The following excitation and detection windows were used to detect fluorescent signal: GFP (Ex 488 nm, Em 500–550 nm), FM-64 (Ex 561 nm, Em 600–650 nm). Images were acquired in 512 × 512 format with a 40X/1.1 water PL APO CS2 objective. Six-day-old seedlings were incubated in ½ MS liquid medium supplemented with 1 μM FM4–64 (Invitrogen) for 5–10 min and washed 3 times in ½ MS before mounting for confocal imaging.

**Protein expression in *Xenopus* oocytes**
DNA constructs encoding AtHv1, PsHv1, SmHv1, and TcHv1, as well as HA-tagged channels and chimeras ChE1-2, ChNC, and ChE3-4.S4, were custom made by GenScript (cDNA sequences of the wild type channels were codon optimized for expression in animal cells). Single amino acid substitutions were introduced by site-directed mutagenesis. Chimeras ChI2-3a, ChI2-3b were prepared using standard PCR techniques. The first twenty residues in the tagged AtHv1 and PsHv1 proteins were MYPYDVPDYASGMNIINTGT and MYPYDVPDYASGMGSGTLTD, respectively. The construct encoding human Hv1 was as in ref. 25. All cDNA sequences were subcloned in the pGEMHE vector between the BamHI and XbaI sites. The resulting plasmids were linearized with NheI or SphI restriction enzymes (New England Biolabs) before in vitro transcription. mRNAs were synthesized using T7 mMessage mMachine transcription kit (Ambion) or HiScribe™ T7 ARCA mRNA Kit (New England Biolabs). All constructs were confirmed by sequencing, and mRNA quality was tested by agarose gel electrophoresis. One to three days before the electrophysiological measurements, mRNAs were injected in *Xenopus* oocytes from Ecocyte Bioscience or Xenopus 1. Injections were performed with a Nanoject II (Drummond Scientific) (50 nl per cell, 0.8–1.5 ng/nl). Cells were kept at 16–18 °C in ND96 medium containing 96 mM NaCl, 2 mM KCl, 1.8 mM CaCl$_2$, 1 mM MgCl$_2$, 10 mM HEPES, 5 mM pyruvate, 50 μg/ml gentamycin (pH 7.2).

**Surface biotinylation and Western blotting**
mRNAs for HA-AtHv1 and HA-PsHv1 were injected in Xenopus oocytes (70 ng per cell). After a 48/60-h incubation at 18 °C in ND96 medium, the cells were transferred in a 12-well plate washed three times with phosphate buffer (PBS) at pH 7.5 and then incubated with 2 mM Sulfo-NHS-Biotin (Thomas Sci.) in PBS for 30 min at room temperature with gentle shaking. In total, 12–20 cells were treated per condition. After quenching the reaction with a solution containing 75 mM NaCl, 25 mM Glycine and 25 mM Tris-HCl at pH 7.5, the cells were washed thrice with PBS, transferred in a micro-centrifuge tube, and then processed as described in ref. 96. Briefly, cells were lysed at 0 °C in 0.5 ml of buffer containing 150 mM NaCl, 1 mM EDTA, 1% v/v NP-40, 1% w/v sodium deoxycholate, 0.1% w/v SDS, protease inhibitor cocktail (Halt, Thermo Sci.) and 10 mM Tris-HCl, pH 7.5. A syringe with a 25-gouge needle was used for breaking up the cells. After separation of cell debris and yolk by centrifugation, the clear lysate was incubated overnight with 40 μl of high-capacity streptavidin agarose resin (Pierce, Thermo Sci.) at 4 °C on a tube rotator. The resin was then washed with 0.5 ml fresh lysis

buffer on ice three times. Following the last wash, captured proteins were released by incubating the resin in 50 μl of 2xLaemmli buffer (containing 10% v/v 2-mercaptoethanol) at room temperature for 20 min, mixing by tapping every 5 min. The supernatant collected after spinning down the resin was subjected to SDS-PAGE (4–20% gradient, Tris-Glycine, Novex, Invitrogen) and Western blotting (Fig. S12b). Following protein transfer, the PVDF membrane (Immobilon-PSQ, Millipore-Sigma) was blocked at 4 °C overnight and then incubated with 250 ng/ml anti-HA monoclonal antibody (2-2.2.14) conjugated with horseradish peroxidase (Invitrogen/Thermo Fisher) for 1.5 h at room temperature. After washing, protein detection was achieved using 1-Step TMB-Blotting reagent (Thermo Sci.). The procedure was performed on two separate batches of oocytes with comparable results.

**Electrophysiology**
Patch clamp measurements were performed in voltage-clamp and inside-out configuration using an Axopatch 200B amplifier controlled by pClamp10 software through an Axon Digidata 1440A (Molecular Devices). The signal was lowpass filtered at 1 kHz (Bessel, −80 dB/decade) before digitalization (2–5 kHz sampling), and further filtered offline at 200 or 150 Hz (Bessel, −80 dB/decade). Leak subtraction and baseline correction were applied after data acquisition. The smallest leak subtracted currents measured before mechanical stimulation ($I_A$) were observed in non-injected oocytes and were in the range 1–2 pA. Examples of leak subtraction are shown in Fig. S12a. Unless otherwise specified, the holding potential was either −60 mV or −80 mV. Voltage ramps used for $I–V$ measurements had rates of 3–4 mV/s. All experiments were carried out at 22 ± 1 °C. Pipettes had 0.8–1.3 MΩ access resistance. Measurements were performed with different combinations of bath and pipette recording solutions. The solution at pH 6.0 contained 100 mM 2-(N-morpholino)ethanesulphonic acid (MES) and 30 mM tetraethylammonium (TEA) methanesulfonate (MS). The solution at pH 5.5 contained 100 mM MES and 40 mM TEA-MS. The solution at pH 6.5 contained 70 mM 1,4-piperazinediethanesulfonic acid and 15 mM TEA-MS. Additionally, all solutions contained 5 mM TEA chloride and 5 mM ethylene glycol-bis(2-aminoethyl)-N,N,N′,N′-tetra-acetic acid. In all cases, the final pH was adjusted with TEA hydroxide. The $\Delta V_{rev}$ measurements used to estimate K$^+$ and Na$^+$ permeabilities relative to H$^+$ were carried out with MES solutions at pH 6.0 in which 1 mM or 10 mM of TEA-MS were replaced with equivalent concentrations of KCl or NaCl. Voltage ramp following the depolarization step had rate of 1 mV/ms. Hv1 inhibitors 2-guanidinobenzimidazole (2GBI) and 5-chloro-2-guanidinobenzimidazole (ClGBI) were from Sigma-Aldrich. All compounds were at the highest purity commercially available.

Changes in bath solution and treatment with Hv1 inhibitors were performed with a computer-controlled gravity-fed multivalve perfusion system (Warner Instruments). Channel inhibition was determined by isochronal current measurements at the end of the depolarization pulses. Mechanical stimulation of membrane patches was carried out either manually with a syringe, measuring the negative pressure on the back of the pipette with a manometer (PM015D, WPI), or with a high-speed pressure clamp (HSPC-1, ALA Scientific) controlled by pCLAMP 10. The duration and magnitude of the stimulus (3 s and −10 mmHg, respectively) were selected to produce maximum priming. Additional stimulations were applied to each membrane patch to determine whether the current could be further increased. Two-electrode voltage clamp measurements were performed 2 days post injection with an Oocyte Clamp OC-725C (Warner Instruments) interfaced to a PC through a 1440A Digidata (Molecular Devices). The bath solution was ND96 and the holding potential was −50 mV. Current traces were filtered at 1 kHz and acquired at 5 kHz. To derive $I_{TEVC}$, the average current from uninjected/control cells was subtracted from the current measured in AtHv1-expressing cells.

## Structural models

Structural models of AtHv1 and PsHv1 were generated by AlphaFold v2.1.0[55] through the AlphaFold2 Colab notebook (AlphaFold2.ipynb). The input sequences comprised residues 51–236 for AtHv1 and residues 70–260 for PsHv1. As a result, the modeled channels started at the S0 helix and did not include the loosely structured N-terminal domain. Models were predicted with the standard version of the program as well as with AlphaFold-multimer[56] but only the structures of the isolated VSDs were considered for analysis. Post-prediction relaxation of the structures in the Amber force field was included[97]. Predicted Local Distance Difference Test (pLDDT) scores[98] for RSN residues in AtHv1 were in the range 70–75. pLDDT scores for the corresponding residues in PsHv1 were in the range 81–83. Pdb files were visualized in PyMOL (Schrödinger, https://pymol.org/2/). Model coordinates for AtHv1 and PsHv1 are provided as Supplementary Data 1 and Supplementary Data 2, respectively.

## Data analysis

Current traces were analyzed using Clampfit10.2 (Molecular Devices) and Origin software (OriginLab). Leak subtraction and rundown correction were carried out as previously described[43]. The $G-V$ relationship in Fig. 2b was derived from tail currents (Fig. 2a). The conductance was determined from $G(V_{test}) = (I_{test} - I_{tail})/(V_{test} - V_{tail})$, where $I_{tail}$ and $V_{tail}$ are the tail current and voltage following the depolarization step at $V_{test}$, and $I_{test}$ is the current measured at the end of the depolarization step. Individual $G-V$ plots were fitted with the Boltzmann equation (Eq. (1)):

$$G(V) = G_{max}/[1 + \exp((V_{1/2} - V)/s)] \qquad (1)$$

where $V_{1/2}$ is the potential of half-maximal activation and $s$ is the slope parameter. Mean $G-V$ relationship was derived from normalization and average of individual $G-V$ plots.

The $G-V$ relationships shown in Figs. S5a, b and S8g were derived from $I-V$ curves using equation:

$$G(V) = I(V)/(V - V_{rev}) \qquad (2)$$

where $V_{rev}$ is the reversal potential of the current (which in Hv channels is $\approx E_H$). $G(V)$ values were then fitted with Eq. (1) and divided by $G_{max}$ for normalization.

$V_T$ values were measures as the $V$-intercept of the linear fit of the steepest segment of the $I-V$ curve. It can be shown that $V_T$ is linked to the $G-V$ relationship by Eq. (3):

$$V_T = (aV_{1/2} + V_{rev})/(a + 1) \qquad (3)$$

where $a = (V_{1/2} - V_{rev})/2s$. This relationship indicates that $V_T$ does not depend on the absolute level of proton current when the $G-V$ is represented by Eq. (1). Figure S5c tests the validity of Eq. (3) in PsHv1 by correlating $V_T$s measured with the $V$-intercept method to $V_T$s calculated form $G-V$s. The V-intercept method can be applied as long as there is enough separation between $V_{rev}$ and $V_{1/2}$. This condition was not met by the $I-V$ of AtHv1 measured at $pH_i = pH_o = 5.5$ (Fig. S8b). So, in that particular case, $V_T$ was calculated from the $G-V$ shown in Fig. S8g.

Derivation of concentration dependence curves for 2GBI and ClGBI were performed as in ref. 51. Concentration dependences were fitted with the Hill equation (Eq. (4)):

$$\%_i = \%_{i, max}[L]^h / \left(IC_{50}^h + [L]^h\right) \qquad (4)$$

where $\%_i$ is the percentage of inhibition at the ligand concentration $[L]$, $\%_{i,max}$ is the percentage of maximal inhibition (assumed to be 100%), $IC_{50}$ is the half-maximal inhibitory concentration, and $h$ is the Hill coefficient.

$\Delta V_{rev}$ values expected for different permeability ratio for $K^+$ and $Na^+$ compared to $H^+$ were calculated using the Goldman-Hodgkin-Katz voltage equation (Eq. (5)):

$$\Delta V_{rev} = (RT/F) \ln ((P_{M^+}[M^+]_o + P_{H^+}[H^+]_o)/(P_{M^+}[M^+]_i + P_{H^+}[H^+]_i)) \qquad (5)$$

with the following ionic concentrations: $[H^+]_i = [H^+]_o = 10^{-6}$ M, $[M^+]_i = 10^{-2}$ M, $[M^+]_o = 10^{-3}$ M ($M^+ = K^+$ or $Na^+$).

Statistical analysis was performed using Origin (OriginLab) and Prism (GraphPad). Data are represented as mean ± SEM, unless otherwise indicated. Datasets with two conditions were compared by applying a Welch's $t$-test. Datasets containing more than two conditions were compared using one-way Welch's ANOVA with Dunnett's T3 multiple comparison test ($p$ values are reported in figure legends). Differences in mean values were considered statistically significant for $p < 0.05$.

## Reporting summary

Further information on research design is available in the Nature Portfolio Reporting Summary linked to this article.

## Data availability

All data supporting the findings of this study are available within the paper and its Supplementary Information. Full image for Fig. S4b is shown in Fig. S12b. Coordinates for structural models shown in Figs. 6 and S10 are provided in two Supplementary Data files. Source data are provided with this paper.

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

## Acknowledgements

The authors wish to thank Jacob Liu and Shane A. C. Wilcox for assistance with DNA construct preparation, K. Bertaux for selecting transgenic plants, and members of the Tombola and De Angeli labs for useful feedback on the manuscript. This work was supported by the National Institute of General Medical Sciences through grant GM098973 to F.T. and by CNRS (ATIP-Avenir grant) and the French National agency of Research (grant Netflux) to A.D.A. We acknowledge the imaging facility MRI, member of the France-BioImaging national infrastructure supported by the French National Research Agency (ANR-10-INBS-04, <<Investments for the future>>), and C. Alcon for assistance.

## Author contributions

F.T. oversaw the project; C.Z., F.T., and A.D.A. designed experiments; C.Z., A.D.A., and P.D.W. performed experiments; C.Z., F.T., and A.D.A. analyzed data; C.Z. and F.T. wrote the first draft of manuscript. All authors reviewed and edited the manuscript.

## Competing interests

The authors declare no competing interests.
