## [Peer Review File · Nature Communications]

Mechanically-primed voltage-gated proton channels from angiosperm plantsREVIEWER COMMENTS

Reviewer #1 (Remarks to the Author):

Hv1 gene is ubiquitously found across species but plant Hv1 has not been characterized. This paper provides the first report of heterologous functional expression of plant Hv1s. Hv1 ortholog (AtHv1) of *Arabidopsis thaliana*, the most popular model species among plants, was characterized in *Xenopus oocyte*. In both two electrode voltage clamp and patch clamp measurement, proton current was not observed from AtHv1. However, after one shot of membrane stretch, authors observed functional Hv1 currents. Authors call this requirement of membrane stretch for appearance of currents “priming”. They also found that other group of plant species showed robust currents without membrane stretch unlike AtHv1. By comparing between AtHv1 and orthologs from plant species which do not require “priming” through performing AI-based method of structural prediction and electrophysiological studies, authors gained insights into mechanisms of stabilized resting state unique to *Arabidopsis* Hv1 channel. The concept of “priming” is novel and interesting. Many force-sensitive ion channels have been identified, but detailed mechanisms by which channel gating is stretch-regulated have been unclear. Findings in this paper will be of impact to researchers who work on stretch sensitive ion channels given that Hv1 is one of the simplest one among stretch sensitive channels. Experiments are carefully done, techniques are sound and the paper is well written.

Major points:

1. I think that ion selectivity of AtHv1 and other plant Hv1 has not been fully tested in this paper. Reversal potential is shifted following Nernst equation with proton selectivity, but authors seem to use medium which does not contain sodium nor potassium in these experiments and therefore it is not solidified whether other cations such as sodium or potassium could permeate through plant Hv1s that they studied (Figure 2 and 3). In particular, since hydrophobic plug residue, phenylalanine on S2, is not conserved in AtHv1, I wonder if ion selectivity may be slightly different from that of other Hv1s.
2. Activation of AtHv1 capabilities by priming with membrane stretch is very clear and deeply studied in this paper. However, process of recovery to silenced state has not been addressed. How long is AtHv1 kept active after priming? In other words, how long does effect of priming continue? I guess that this depends on how long inside out patch measurement is possible. Any information of experiments with longer time recording would be included as either data or sentences.
3. Authors quantitatively estimated “priming” effect by dividing current magnitude of post “priming” channel against pre-“priming” channel (as I_B/I_A). This calculation is based on the idea that outward current is small but not zero before membrane stretch, because zero current will make I_B/I_A infinite. This seems paradoxical, since “priming” idea comes from that channel is completely silenced without priming and stretch manipulation wakes the channel so that channel is ready to be voltage-gated. In fact, authors mentioned in the first page of Results that they did not see proton current either inside out patch or TEVC recording, which contradicts with raw traces before membrane stretch in Fig4b. I speculate that small but clear outward current before priming in Fig4b could be due to some membrane stretch episodes during formation of inside out patches. If so, there could be some cases where there are no outward current (zero current before priming) which causes infinite value of I_B/I_A . In addition, this style of quantification will be much biased by leak current (how cell is healthy and also how leak subtraction is performed) of cells before “priming”. These apparent discrepancies and ambiguity of quantitative estimation need to be addressed in the paper.

4. In the protocol, priming was done by negative pressure. What happens if reverse pressure (positive pressure) is applied?
5. Authors previously showed human Hv1 is stretch sensitive. I wonder if stretch sensitivity is more sensitive in many plant Hv1s than in mammalian Hv1, or only AtHv1 and TcHv1 have remarkable stretch sensitivity. Was PsHv1 more stretch sensitive than human Hv1? Figure 4c seems to show that there is no significant difference between human Hv1 and PsHv1, but this was only done with a fixed protocol.

Minor points:

6. From the pulse protocol, leak subtraction (P over N) does not seem to be done. This should be clarified in Figure legend or Methods.
7. In Fig4b, what does inward current upon negative pressure represent in *T. cacao*? Was this a particular case or did authors repeatedly see such inward currents? Explanation for inward current would be included in the results or the figure legend.
8. On Line 323 (On page 13), "octopus" is not correct. It is *Aplysia* (sea slug).
9. Membrane stretch dependence of channel gating may depend on lipid compositions. Therefore, experiments in a different heterologous expression system such as mammalian cells will strengthen the conclusions. If such test is performed, statement would be included (probably as data in supplementary figure or in the text).
10. In Fig4c legend, please include information of species abbreviations of *H. s.* (probably *Homo sapiens*).

Reviewer #2 (Remarks to the Author):

This is a well organized and well written manuscript about a novel mechanism of regulation of voltage-gated proton Hv1 channels. The authors show that one group of plants have Hv1 channels that need to be mechanically primed before they can be activated by voltage. The data is clear and the residues involved are identified by the authors through molecular modelling and mutagenesis. I think this would be of great interest in the field of ion channels. I only have a few comments and suggestions.

1. The authors show that the pH shifts are larger in these plant Hv1 channels than in mammalian Hv1 channels. They authors suggests that this means that an additional pH sensor is needed in addition to the previously proposed pH sensing voltage sensor. However, one would expect that pH shifts would be bigger with shallower slopes of GVs, as is the case for these plant channels. This can be seen if one adds ΔG to the exponent of the Boltzmann and rearranges this to a $\Delta V_{0.5}$: $\exp(-z(V-V_{0.5})+\Delta G) = \exp(-z(V-V_{0.5} + \Delta V_{0.5})) \dots \Delta V_{0.5} = \Delta G/z$. so if z is smaller then $\Delta V_{0.5}$ will be larger for the same ΔG .
2. It is not clear to me what the physiological function of this mechanical stimuli would be? Would be nice to more explicitly couple mechanical stimuli in these plants to opening of the channel and the resulting physiological effect of the H^+ flux...?
3. what is the role of inward currents in some plant Hv1 channels? This would seem detrimental to the plants.. please discuss..

Reviewer #3 (Remarks to the Author):

This complex paper by Zhao et al. brilliantly demonstrates that the proton channel of the angiosperm plant *Arabidopsis thaliana* is electrically silent in the absence of a mechanical stimulation. The authors suggest that the molecular mechanisms underlying the recovery of the voltage- and pH-dependent activation of AtHv1 is mediated by a sort of mechanical “priming”.

The properties of a non-silent Hv channel from a non-flowering plant (namely *Picea sitchensis*) have also been characterized and compared with those of homologous non-plant proton channels.

A series of AtHv1 and PsHv1 chimeras was able to show that AtHv1 may recover the activation and supports the conclusions that the major determinants of mechanical priming of AtHv1 channel are located in the ring-shaped networks as well as in the residues at the top of the S4 segment.

These results will be useful not only for understanding the modulation of other Hv plant channels, but also for the regulation of homologous proteins from the animal and fungi kingdom.

The paper is complete and also very precise and articulated both in the technical aspects as well as the more general comments. It will possibly open new perspectives in the field of proton transport in the plant field as well as on the consequences of proton accumulation both in the areas of human health and agricultural production.

The manuscript is well documented with citations of relevant and recent papers in the area of interest as one would expect by the two corresponding authors who are leaders in the fields of ionic transport in plants and proton channels, respectively.

The work fully supports the conclusions of the authors and no additional evidences are needed.

I don't see major flaws in the analysis, in the interpretation of the data and in the conclusions: the article is well written and convincing. The data analysis is very complete and well summarized in 8 accurate figures plus 10 supplementary figures.

The methodologies are very advanced, based on well-established techniques e. g. electrophysiology applied to protein/channel expression in oocytes, confocal fluorescence microscopy, molecular biology, as well as bioinformatics to generate (by AlphaFold2) structural models of the channels and chimeras of interest.

The experiments are detailed and provide sufficient information for their reproduction by other researchers specifically interested in this topic or in the more general aspects.

I have only a few minor comments or suggestions:

- 1) In supplementary figure S9, the representation of the structural model of AtHv1 and PsHv1 (a and b) are inverted. See the text (lines 261-265) and correspondent legend of figure S9.
- 2) Furthermore, in figure S9, I suggest to add O (for Outside) and I (for Inside) similarly to what reported in other figures where structural models are illustrated (e. g. Fig. 5, 7, ...)
- 3) Line 266: Ch3-4.S4K, a dot between 4 and K is missing? Compare with Ch3-4.S4.K at line 246 and Figure 5a.
- 4) Since the dimensions of some characters are definitely very small (for example see Fig.5

ChE1-2, ChI2-3a/b, ...), if the size of these figures will be further reduced in the final publication, I am concerned that they will become almost unreadable when printed.

We would like to thank the Reviewers for their helpful assessment of our manuscript. We provide detailed answers to their remarks below (text in blue). Major revisions of the text (manuscript + SI) are colored in red in the marked-up file.

Reviewer #1 (Remarks to the Author):

Hv1 gene is ubiquitously found across species but plant Hv1 has not been characterized. This paper provides the first report of heterologous functional expression of plant Hv1s. Hv1 ortholog (AtHv1) of *Arabidopsis thaliana*, the most popular model species among plants, was characterized in *Xenopus oocyte*. In both two-electrode voltage clamp and patch clamp measurement, proton current was not observed from AtHv1. However, after one shot of membrane stretch, authors observed functional Hv1 currents. Authors call this requirement of membrane stretch for appearance of currents “priming”. They also found that other group of plant species showed robust currents without membrane stretch unlike AtHv1. By comparing between AtHv1 and orthologs from plant species which do not require “priming” through performing AI-based method of structural prediction and electrophysiological studies, authors gained insights into mechanisms of stabilized resting state unique to *Arabidopsis* Hv1 channel. The concept of “priming” is novel and interesting. Many force-sensitive ion channels have been identified, but detailed mechanisms by which channel gating is stretch-regulated have been unclear. Findings in this paper will be of impact to researchers who work on stretch sensitive ion channels given that Hv1 is one of the simplest one among stretch sensitive channels. Experiments are carefully done, techniques are sound and the paper is well written.

Major points:

1. I think that ion selectivity of AtHv1 and other plant Hv1 has not been fully tested in this paper. Reversal potential is shifted following Nernst equation with proton selectivity, but authors seem to use medium which does not contain sodium nor potassium in these experiments and therefore it is not solidified whether other cations such as sodium or potassium could permeate through plant Hv1s that they studied (Figure 2 and 3). In particular, since hydrophobic plug residue, phenylalanine on S2, is not conserved in AtHv1, I wonder if ion selectivity may be slightly different from that of other Hv1s.

As suggested by the reviewer, we performed new measurements to test whether sodium or potassium can permeate plant Hv channels and show the results in a new supplemental figure (Fig. S8) discussed in revised text (line 207 – 216). We recorded currents from PsHv1 and AtHv1 in the presence and absence of transmembrane gradients of sodium or potassium under symmetrical pH conditions and measured the resulting shifts in reversal potential (ΔV_{rev}). The results indicate that the permeabilities of sodium and potassium through these channels is negligible (P_{Na^+}/P_{H^+} and $P_{K^+}/P_{H^+} < 10^{-5}$) (Fig. S8b), as previously observed with other Hv proteins. In addition, the Nernstian dependency of the reversal potential on the proton gradient (Fig. 2c-d and Fig. S7) was measured under conditions of symmetrical chloride concentrations, ruling out this anion as a significant contributor to the measured current. We now mention this in the revised text (lines 209 – 211).

2. Activation of AtHv1 capabilities by priming with membrane stretch is very clear and deeply studied in this paper. However, process of recovery to silenced state has not been addressed. How long is AtHv1 kept active after priming? In other words, how long does effect of priming continue? I guess that this depends on how long inside out patch measurement is possible. Any information of experiments with longer time recording would be included as either data or sentences.

After priming, AtHv1 remained active for the duration of the measurements, which were usually completed within 5 to 10 minutes. The return to the silent state after priming could not be followed over time due to superimposing rundown of the current, which was observed in AtHv1 and the other plant proton channels and resembled the equivalent process previously reported in human Hv1 (Tombola et al. *Neuron* 2008). We

now provide this information in the revised text (lines 199 – 202). The rundown process is distinct from the return to the silent state as mechanical stimulation after rundown failed to recover the current.

3. Authors quantitatively estimated “priming” effect by dividing current magnitude of post “priming” channel against pre-“priming” channel (as I_B/I_A). This calculation is based on the idea that outward current is small but not zero before membrane stretch, because zero current will make I_B/I_A infinite. This seems paradoxical, since “priming” idea comes from that channel is completely silenced without priming and stretch manipulation wakes the channel so that channel is ready to be voltage-gated. In fact, authors mentioned in the first page of Results that they did not see proton current either inside out patch or TEVC recording, which contradicts with raw traces before membrane stretch in Fig4b. I speculate that small but clear outward current before priming in Fig4b could be due to some membrane stretch episodes during formation of inside out patches. If so, there could be some cases where there are no outward current (zero current before priming) which causes infinite value of I_B/I_A . In addition, this style of quantification will be much biased by leak current (how cell is healthy and also how leak subtraction is performed) of cells before “priming”. These apparent discrepancies and ambiguity of quantitative estimation need to be addressed in the paper.

The reviewer is correct in stating that “...outward current before priming in Fig4b could be due to some membrane stretch episodes during formation of inside out patches.” This is somewhat unavoidable because some suction needs to be applied to the pipette to generate the membrane seal. We would like to clarify that leak subtraction and baseline correction were applied after data acquisition. The smallest leak subtracted currents measured before mechanical stimulation (I_A) were observed in non-injected oocytes and were in the range 1 – 2 pA. In the revision, this information is reported in lines 531 – 534. We also added a supplemental figure (Fig. S12a) with examples of leak subtractions of proton currents from membrane patches containing PsHv1 and AtHv1. The figure compares the raw traces to the leak subtracted traces before priming to show impact of leak on total current.

4. In the protocol, priming was done by negative pressure. What happens if reverse pressure (positive pressure) is applied?

We are aware of work on the mechanically gated channel Piezo1 in which positive pressure was successfully applied to activate the channel (Lewis & Grandl eLife 2015). However, the inside out patches we need to use for plant Hv channels are much larger and closer to the mouth of the pipette than those used to study Piezo1. As a result, application of positive pressure resulted in loss of seal integrity. For Piezo1, it was concluded that lateral membrane tension and not membrane curvature activates the channel. Since we could not measure the effect of positive pressure on Hv priming, we did not make any conclusions about membrane curvature.

5. Authors previously showed human Hv1 is stretch sensitive. I wonder if stretch sensitivity is more sensitive in many plant Hv1s than in mammalian Hv1, or only AtHv1 and TcHv1 have remarkable stretch sensitivity. Was PsHv1 more stretch sensitive than human Hv1? Figure 4c seems to show that there is no significant difference between human Hv1 and PsHv1, but this was only done with a fixed protocol.

In this first study of plant Hvs, we focused on the maximal increase in current produced by the mechanical stimulus, as the feature was sufficiently robust for mechanist studies. We found that this maximal increase, quantified as I_B/I_A , was similar for PsHv1 and human Hv1 and much larger for AtHv1 and TcHv1. To determine stretch sensitivity, one would have to measure the current increase as a function of the intensity of mechanical stimulation. This was previously done for several mechanically gated channels and we plan to do similar studies on the priming of Hv channels in the future. However, we should note that priming is not quickly reversible, so each consecutive stimulation with increased intensity acts additively on top of

previous stimulations, which means that we cannot use the same approaches used for Piezo1 and other mechanically gated channels to measure stretch sensitivity.

Minor points:

6. From the pulse protocol, leak subtraction (P over N) does not seem to be done. This should be clarified in Figure legend or Methods.

Correct, as mentioned in answer to point 3, leak subtraction was performed post-acquisition in all cases. We clarify this in the revised Methods section and show examples of leak subtraction in Fig. S12a.

7. In Fig4b, what does inward current upon negative pressure represent in *T. cacao*? Was this a particular case or did authors repeatedly see such inward currents? Explanation for inward current would be included in the results or the figure legend.

We have observed the inward current during the application of the negative pressure pulse routinely, not just in the case shown in Fig. 4b. Even though it seems to be larger with TcHv1, we have seen it with other plant Hvs as well, but not with animal or fungal Hvs. While it is tempting to conclude that the inward current is directly mediated by plant Hv channels, or at least some of them, we suspect a more indirect mechanism given the lack of correlation between the size of the current and the number of channels present in the patch (evaluated by the size of the current elicited by membrane depolarization after priming). We now discuss the inward current in the legend for Fig. 4b, as suggested by the reviewer.

8. On Line 323 (On page 13), "octopus" is not correct. It is *Aplysia* (sea slug). We have amended the text accordingly. Thank you.

9. Membrane stretch dependence of channel gating may depend on lipid compositions. Therefore, experiments in a different heterologous expression system such as mammalian cells will strengthen the conclusions. If such test is performed, statement would be included (probably as data in supplementary figure or in the text).

Our goal was to compare the functional properties of different plant Hv channels under the same conditions in a heterologous system that is commonly used to study plant channels and transporters. We agree with the reviewer that the dependence on lipid composition of plant Hv activity warrants further investigation. However, our attempts to express these channels in HEK293 cells resulted in sick cells that produced leaky patches. As a result, we are evaluating other expression systems for future studies.

10. In Fig4c legend, please include information of species abbreviations of *H. s.* (probably *Homo sapiens*).

Done.

Reviewer #2 (Remarks to the Author):

This is a well organized and well written manuscript about a novel mechanism of regulation of voltage-gated proton Hv1 channels. The authors show that one group of plants have Hv1 channels that need to be mechanically primed before they can be activated by voltage. The data is clear and the residues involved are identified by the authors through molecular modelling and mutagenesis. I think this would be of great interest in the field of ion channels. I only have a few comments and suggestions.

1. The authors show that the pH shifts are larger in these plant Hv1 channels than in mammalian Hv1 channels. They authors suggests that this means that an additional pH sensor is needed in addition to the previously proposed pH sensing voltage sensor. However, one would expect that pH shifts would be bigger with shallower slopes of GVs, as is the case for these plant channels. This can be seen if one adds ΔG to

the exponent of the Boltzmann and rearranges this to a $\Delta V_{0.5}$: $\exp(-z(V-V_{0.5})+\Delta G) = \exp(-z(V-V_{0.5} + \Delta V_{0.5}))$... $\Delta V_{0.5} = \Delta G/z$..so if z is smaller then $\Delta V_{0.5}$ will be larger for the same ΔG ..

Yes, the shifts in the voltage range of activation of plant Hv channels as a function of ΔpH appear larger than those measured in animal Hvs. However, when the effects of changes in absolute pH are taken into account, the shifts are not much different. We thank the reviewer for pointing out the alternative relationship between GV slope and ΔpH sensitivity. We have now revised the discussion of our original point to this: "The lack of the first charged residue on the S4 motif of plant Hvs raised the possibility that the ΔpH dependence of these channels could be substantially different compared to other Hvs. However, this was not the case, as the voltage range of activation of PsHv1, AtHv1, and SmHv1 shifted 40 – 67 mV per ΔpH unit (values corrected for pH dependence under symmetrical conditions). These observations suggest the existence of redundancies in the mechanism of ΔpH sensing in Hv proteins." The revised sentence focuses more on the lack of a substantial difference rather than the direction of the difference.

2. It is not clear to me what the physiological function of this mechanical stimuli would be? Would be nice to more explicitly couple mechanical stimuli in these plants to opening of the channel and the resulting physiological effect of the H⁺ flux...?

We agree with the reviewer and introduced a more explicit paragraph on the potential physiological function of AtHv1 and on its mechanical stimulation (lines 398 – 412).

"Plants cells experience mechanical stresses induced by the environment and during the development of tissues and organs (77-79). The cell-wall and the plasma membrane are the two major actors for the perception and the responses to mechanical stimulations (80). Mechanosensitive channels are present in the plasma membrane of plant cells (81) and the cell wall composition is modified by mechanical stimulation (82). In the root vasculature, where AtHv1 is expressed, the composition of the cell wall is modified during the elongation and secondary cell wall deposition (SCW) phases, resulting in changes of extensibility and stiffness (83). Recent results show that anisotropic stress distribution in the root vasculature defines the patterning of vascular cells controlling their development (84,85). It is known that during elongation and SCW there is a production of ROS, and AtHv1 could participate to regulate such oxidative bursts (86), similarly to its mammalian homologue in phagocytic cells (1). In this context, the mechanical forces experienced by the plasma membrane in root vascular cells could activate or deactivate AtHv1. Because in plant cells the membrane potential is largely dependent on the pH gradient (87,88) the activation of AtHv1 could cause changes in membrane potential and consequently act on intracellular signaling. Of particular interest is the signaling involved in auxin-dependent cell growth given its known dependence on H⁺ transport across the plasma membrane (89)."

3. what is the role of inward currents in some plant Hv1 channels? This would seem detrimental to the plants.. please discuss..

The transmembrane pH gradient and membrane potential in plant cells is maintained by the activity of powerful proton pumps, which constantly move protons out of the cell. It is possible that some plant Hv1 channels become activated under conditions that favor inward proton flux, but the activation would be likely transient due to the counteracting action of the pumps. Although brief, the opening of the Hv1 channels could be sufficient to initiate intracellular signaling via transient change in membrane potential and/or local pH.

Reviewer #3 (Remarks to the Author):

This complex paper by Zhao et al. brilliantly demonstrates that the proton channel of the angiosperm plant *Arabidopsis thaliana* is electrically silent in the absence of a mechanical stimulation. The authors suggest

that the molecular mechanisms underlying the recovery of the voltage- and pH-dependent activation of AtHv1 is mediated by a sort of mechanical “priming”.

The properties of a non-silent Hv channel from a non-flowering plant (namely *Picea sitchensis*) have also been characterized and compared with those of homologous non-plant proton channels.

A series of AtHv1 and PsHv1 chimeras was able to show that AtHv1 may recover the activation and supports the conclusions that the major determinants of mechanical priming of AtHv1 channel are located in the ring-shaped networks as well as in the residues at the top of the S4 segment.

These results will be useful not only for understanding the modulation of other Hv plant channels, but also for the regulation of homologous proteins from the animal and fungi kingdom.

The paper is complete and also very precise and articulated both in the technical aspects as well as the more general comments. It will possibly open new perspectives in the field of proton transport in the plant field as well as on the consequences of proton accumulation both in the areas of human health and agricultural production.

The manuscript is well documented with citations of relevant and recent papers in the area of interest as one would expect by the two corresponding authors who are leaders in the fields of ionic transport in plants and proton channels, respectively.

The work fully supports the conclusions of the authors and no additional evidences are needed.

I don't see major flaws in the analysis, in the interpretation of the data and in the conclusions: the article is well written and convincing. The data analysis is very complete and well summarized in 8 accurate figures plus 10 supplementary figures.

The methodologies are very advanced, based on well-established techniques e. g. electrophysiology applied to protein/channel expression in oocytes, confocal fluorescence microscopy, molecular biology, as well as bioinformatics to generate (by AlphaFold2) structural models of the channels and chimeras of interest.

The experiments are detailed and provide sufficient information for their reproduction by other researchers specifically interested in this topic or in the more general aspects.

I have only a few minor comments or suggestions:

1) In supplementary figure S9, the representation of the structural model of AtHv1 and PsHv1 (a and b) are inverted. See the text (lines 261-265) and correspondent legend of figure S9.

We thank the reviewer for noticing this confusing point. The mistake was in the text referring to Fig. S9 (now renumbered as Fig. S10) where a and b were inverted. The text has been revised accordingly (lines 277-278).

2) Furthermore, in figure S9, I suggest to add O (for Outside) and I (for Inside) similarly to what reported in other figures where structural models are illustrated (e. g. Fig. 5, 7, ...)

Done.

3) Line 266: Ch3-4.S4K, a dot between 4 and K is missing? Compare with Ch3-4.S4.K at line 246 and Figure 5a.

Done.

4) Since the dimensions of some characters are definitely very small (for example see Fig.5 ChE1-2, ChI2-3a/b, ...), if the size of these figures will be further reduced in the final publication, I am concerned that they will become almost unreadable when printed.

We increased the fonts of the labels in Fig. 5 and in other figures where they were particularly small. Thanks.

REVIEWERS' COMMENTS

Reviewer #1 (Remarks to the Author):

In this version all concerns which I raised were addressed fully and I do not have further concern. This work is of high significance in terms of molecular and functional properties of Hv1 channel.

Reviewer #2 (Remarks to the Author):

The authors have responded well to my comments and suggestions.

Reviewer #3 (Remarks to the Author):

I am fully satisfied with the answers of the authors to my suggestions